# Time series of alpine snow surface radiative temperature maps from high precision thermal infrared imaging

Sara Arioli[1], Ghislain Picard[1], Laurent Arnaud[1], Simon Gascoin[2], Esteban Alonso-González[3], Marine Poizat[1], and Mark Irvine[4]

[1]Univ. Grenoble Alpes, CNRS, INRAE, IRD, Grenoble INP, IGE, 38000 Grenoble, France
[2]Centre d'Etudes Spatiales de la Biosphère, Université de Toulouse,CNRS/CNES/IRD/INRA/UPS, Toulouse, France
[3]Instituto Pirenaico de Ecología, Consejo Superior de Investigaciones Científicas, Jaca, Spain
[4]ISPA, INRAE, Bordeaux Sciences Agro, F-33140 Villenave d'Ornon, France

**Correspondence:** Sara Arioli (sara.arioli@univ-grenoble-alpes.fr)

**Abstract.** The surface temperature of snow cover is a key variable, as it provides information about the current state of the snowpack, helps predict its future evolution, and enhances estimations of the snow water equivalent. Although satellites are often used to measure surface temperature and despite the difficulty to retrieve accurate surface temperature from space, calibration-validation datasets over snow-covered areas are scarce. We present a dataset of extensive measurements of the radiative surface temperature of snow acquired with an uncooled Thermal Infrared (TIR) camera. The set accuracy goal is 0.7 K, which is the radiometric accuracy of the TIR sensor of the future CNES/ISRO TRISHNA mission. TIR images have been acquired over two winter seasons, November 2021 to May 2022 and February to May 2023 at the Col du Lautaret, 2057 m a.s.l. in the French Alps. During the first season, the camera operated in the off-the-shelf configuration, with a rough thermal regulation (7°C - 39°C). An improved setup with a stabilized internal temperature was developed for the second campaign, and comprehensive laboratory experiments were carried out in order to characterize the physical properties of the TIR camera components and its calibration. A thorough processing including radiometric processing, orthorectification and a filter for poor visibility conditions due to fog or snowfall have been performed. The result is two winter season timeseries of 130,019 maps of the surface radiative temperature of snow with meter-scale resolution over an area of 0.5 $km^2$. The validation is performed against precision TIR radiometers. We found an absolute accuracy (MAE) of 1.28 K during winter 2021-2022 and 0.67 K for spring 2023. The efforts to stabilize the internal temperature of the TIR camera therefore led to a notable improvement of the accuracy. Although some uncertainties persist, particularly the temperature overestimation during melt, this dataset represents a major advance in the capacity to monitor and map surface temperature in mountainous areas, and to calibrate-validate satellite measurements over snow-covered areas of complex topography. The complete dataset is at https://doi.org/10.57932/8ed8f0b2-e6ae-4d64-97e5-1ae23e8b97b1 (Arioli et al., 2024a) and https://doi.org/10.57932/1e9ff61f-1f06-48ae-92d9-6e1f7df8ad8c (Arioli et al., 2024b).

# 1 Introduction

Snow is a crucial element of alpine ecosystems, as its high reflectivity cools the climate of high-altitude environments, while its insulating properties contribute to preserve glaciers (Barry, 2002). In alpine areas, it represents a crucial storage of water for the downstream ecosystems and human population (Grabherr et al., 2010). Indeed, with over one sixth of the world population relying on the high-mountain cryosphere as a water supply (Barnett et al., 2005), extensive monitoring of the snowpack is key to improving the understanding of future change in water resources (Fayad et al., 2017; Beniston et al., 2018).

The surface temperature of snow is one of the key variables to monitor the status of the snowpack. It plays a major role in the surface energy budget, that describes the exchanges of energy between the snowpack and the atmosphere (Armstrong and Brun, 2008), it provides insights about the onset and duration of surface melt (Alonso-González et al., 2023), and it influences the evolution of the microstructural and optical properties of the snow grains (Colbeck, 1989; Flanner and Zender, 2006). However, surface temperature variations are difficult to capture and predict. Indeed, due to the insulating properties of snow, thermal inertia is low, allowing minute-scale changes of the surface temperature. Furthermore, in mountainous areas it varies largely in space, at meter and longer scales, because of the complex terrain (Lundquist and Cayan, 2007). Indeed, phenomena such as 1) variable illumination intensity according to surface slope and orientation, 2) reflection of sunlight and emission of thermal infrared radiation towards neighboring slopes, 3) variations of the air temperature with altitude due to the lapse rate and atmospheric turbulence, induce large spatial variations in the local surface energy budget (Fierz et al., 2003; Adams et al., 2011; Robledano et al., 2022).

Thermal infrared (TIR) sensors on board satellites and other spatial platforms, such as NASA's Landsat 8/9 TIRS/2, ECOSTRESS, ASTER and NOAA's AVHRR monitor the surface temperature of snow-covered areas extensively and regularly (Roy, 2014; Hook and Hulley, 2019; Hulley and Hook, 2010; Kerr et al., 1992). Over mountainous terrain, however, the resolution of most sensors, which is in order of 100 m to 1 km, is coarse with respect to the spatial temperature variations within a single pixel, leading to higher measurement uncertainty (Hall et al., 2008; Simó et al., 2016; He et al., 2019). Also, the revisit time of the TIR sensors currently in orbit, which is in the order of half a day for sensors at 1 km resolution and up to 16 days for ASTER at 90 m resolution, combined with the intermittency of cloud cover, makes the temporal distance between successive useful clear-sky images significantly larger than the typical timescales of the surface temperature variation (Wang et al., 2001). A new generation of thermal infrared sensors are to be launched between 2026 and 2030 on board the TRISHNA, LSTM and SBG satellites (Buffet et al., 2021; Bernard et al., 2023; Stavros et al., 2023). These sensors will have an enhanced resolution between 50 m and 70 m, a revisit of $\leq$3 days individually, achieving 1 day when combined (day pass). They will therefore increase significantly the amount of high-resolution thermal data available over terrestrial surfaces, developing the current understanding of the snow surface energy budget and the estimation of water resources in Alpine areas and cold regions (Alonso-González et al., 2023). However, some difficulties inherent to remote measurements in mountainous areas remain to be solved. Indeed, most atmospheric correction algorithms of satellite images only partially account for the underlying topography (Zhu et al., 2020). Additionally, how the sub-pixel variability of the snow surface temperature affects satellite measurements is still unclear. These uncertainties result from the scarcity of calibration-validation initiatives of thermal infrared acquisition over snow

covered areas (Dybkjær et al., 2012; Høyer et al., 2017). In addition, because of its extent and well-known temperature of 0°C during the melting season, snow is an interesting material for the scope of calibration-validation.

Recently, uncooled thermal infrared cameras have become affordable means to perform high resolution and high revisit time measurements of the surface temperature, similarly to webcams. However, their use has been limited by multiple instrumental biases that make these instruments far from an off-the-shelf solution for scientific applications requiring high accuracy (Aasen et al., 2018). Indeed, measurements are extremely sensitive to the temperature of the sensor itself and of the camera body (Budzier and Gerlach, 2015), and the instrument response drifts over time (Olbrycht et al., 2012). This causes offsets of several K on the measured temperature and creates artifacts in the image (Riou et al., 2004). Many of these issues are partially corrected by the "Flat Field Correction" (FFC). During the FFC, a shutter of known temperature and emissivity is regularly measured by the sensor. The offset between the measurement and the real shutter temperature is then applied to the following measurements, compensating for the errors that build up during operation, pixel by pixel (Nugent et al., 2013; Virtue et al., 2021). The FFC is typically applied every few seconds to few minutes. Still, even with a FFC system, most manufacturer's declared accuracy of TIR cameras ranges between 2 and 5 K (Kelly et al., 2019). This is insufficient for the purpose of TIR satellite calibration-validation, considering that most space-borne TIR sensors have a radiometric accuracy between 0.1 K and 0.8 K (Barsi et al., 2014; Smith et al., 2020; Smyth and Logan, 2022). As a result, the attempts to acquire thermal imagery on snow surfaces in scientific literature are scarce and mainly focus on temperature anomalies rather than on absolute surface temperature (Shea et al., 2012; Kraaijenbrink et al., 2018; Wigmore and Molotch, 2023).

This study presents observations of the surface temperature over 2 winter seasons in an Alpine catchment with complex terrain (Col du Lautaret in the French Alps). The dataset includes high frequency (2 min) timeseries of images of the surface temperature acquired with an uncooled TIR camera. The particularity of this dataset is the high quality of the calibration and stabilization of the camera, and the availability of ancillary measurements to perform a precise cross-evaluation with the aim to achieve high accuracy. The accuracy goal fixed for the dataset is 0.7 K, which is the radiometric accuracy expected for the TRISHNA satellite, that carries the precursor of the new generation of TIR imaging radiometers. This paper describes the thorough step-by-step processing of raw data and the careful assessment of the measurement uncertainty. During processing, snow is treated as a black body. While the actual emissivity of snow is between 0.96 and 0.99 at nadir (Hori et al., 2006), choosing a precise value for snow emissivity is challenging because of its potential variations with grain types and presence of surface water during the melting season (Dozier and Warren, 1982; Hori et al., 2014). Other in-situ measurements include snow surface temperature from TIR radiometers deployed in the camera's field of view (FOV) – used as ground truth –, RGB images of the scene, timeseries of the internal temperature of the TIR camera and meteorological parameters that are helpful to the interpretation of the camera measurements, such as incoming longwave radiation, air temperature and humidity, and wind speed. The study site is described in Section 2. The instruments used, the lab characterization of the TIR camera and the processing of the measurements are described in Section 3, 4 and 5 respectively. The results are presented in Section 6, validated in Section 7 and discussed in Section 8.

## 2 Study site

The two winter measurement campaigns took place at the Col du Lautaret site in the French Alps (Fig. 1, 45.0347 N, 6.4051 E),
at an altitude of 2057 m a.s.l. The imaged site is located next to the Col du Lautaret pass, with the TIR camera pointing West.
It includes a North facing slope (Fig. 2, left) and a South facing slope (Fig. 2, right) and spans altitudes between 1950 m and
2200 m. The Meije massif in the Écrins mountain range is visible in the background. South of the col, the relief steeply climbs
above 3000 m, masking sunlight on the North facing slope of the pass during most of winter. Otherwise, the South facing
slope receives at least some sunlight everyday. Snow covers the soil approximately between December and April every winter
with very high spatial variability in snow depth. Advanced meteorological parameter are measured continuously at the Fluxalp
ICOS station, 500 m North of the col, and are available in open access (https://meta.icos-cp.eu/resources/stations/ES_FR-CLt).

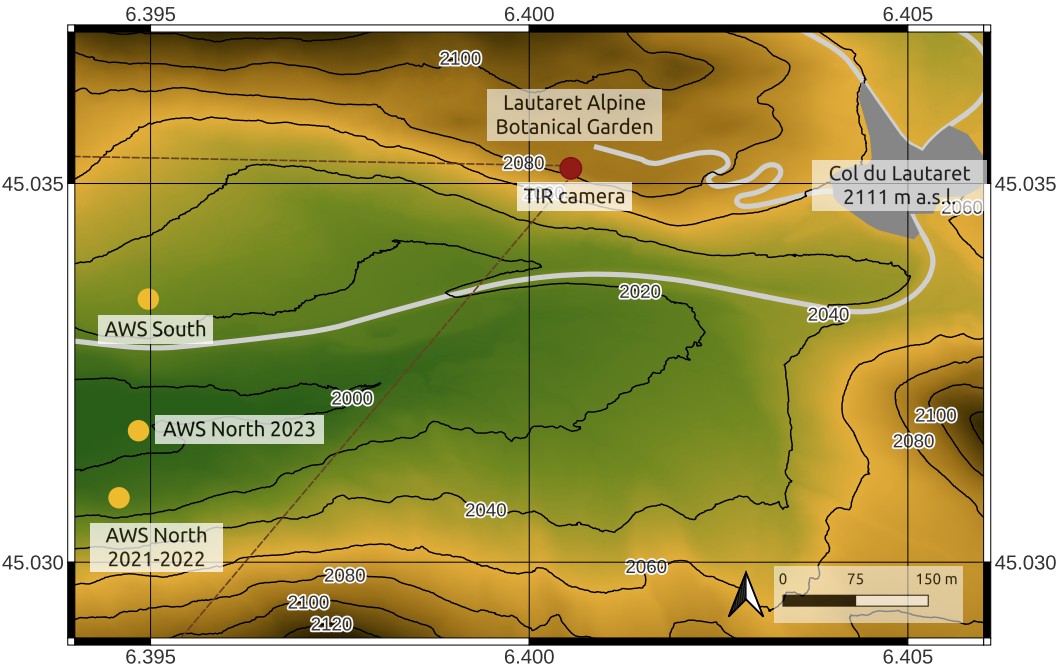

**Figure 1.** Map of the West side of the Col du Lautaret, France (45.0347°N, 6.4051°E) with the location of the TIR camera, its field of view
and two Automatic Weather Stations deployed on the North and South faces. The AWS-North was relocated between winter 2021-2022 and
spring 2023.

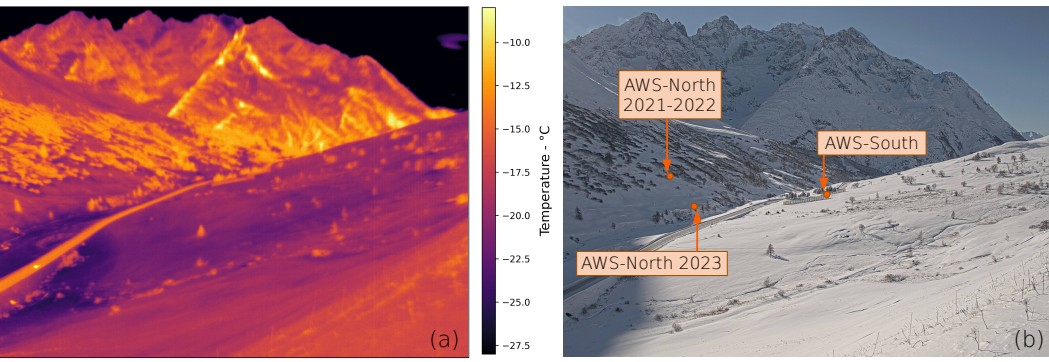

**Figure 2.** Photographs taken by the TIR camera (a) and a standard RGB camera (b).

## 3 Instruments

### 3.1 Thermal Infrared camera

The timeseries of surface temperature images are acquired using an Optris Pi640 camera. Its sensor consists of a focal plane array of 640x480 microbolometers that measure the incoming radiance in the thermal infrared domain, between 8 $\mu$m and 14 $\mu$m (Opt, 2020). Microbolometers work at ambient temperature and therefore do not require expensive cooling systems. However, their response is unstable in time and particularly sensitive to environmental temperature changes. To compensate for these two sources of uncertainty, the camera comes with an internal shutter system to perform FFC. Besides its radiometric utility, the FFC also reduces noise across the image. The FFC process is controlled by the camera software. In addition, the camera is enclosed in an aluminum casing with a basic thermal regulation system (between 10°C and 30°C). The conversion of the signal from digital numbers to temperature is also performed by the camera software. The measurement output is an array of 640x480 temperatures at a resolution of 0.1 K, recorded as a comma-separate-value (csv) file. We configured the acquisition to occur every 2 minutes during field campaigns.

This off-the-shelf setup of the TIR camera is used for the whole 2021-2022 winter campaign, between November 2021 and May 2022. However, we identified that the large internal fluctuations of temperature jeopardized our goal in terms of accuracy. For spring 2023, the setup was therefore modified. The camera was isolated inside the original casing, in a small enclosure whose temperature was kept close to constant (<2K fluctuation) by a thermoelectric module (TEM), while the rest of the casing was used to evacuate the heat excess (Fig. 3a). The volume with the camera and the rest of the casing was both ventilated for the temperature to be uniform inside. Because of the limited power of the TEM, the presence of active electronics inside the camera body and wide external temperature variations, the internal temperature was typically between -0.5 K to +1.0 K with respect to the TEM settings during operation, but varying slowly over time.

The camera in its casing, paired with a waterproof box containing both camera and TEM electronics, was installed on the South wall of the museum building of the Lautaret garden (6.4006 E, 45.0352 N, Fig. 3b). A small wooden roof was

installed on top of the camera to avoid overheating due to direct exposition to sunlight and accumulation of snowfall. The internal temperature of the camera was initially set to 15°C. However, this temperature proved to be too low with respect to the limited heat evacuation in the rear of the casing of this setup, causing significant overheating when the external temperature approaches 15°C. It was then raised to 22°C on the 2023-02-17 after several episodes of overheating occurred during the first week of measurements. By raising the TEM setting temperature from 15°C to 22°C, the required heat evacuation was reduced and no overheating episodes happened during the rest of the season.

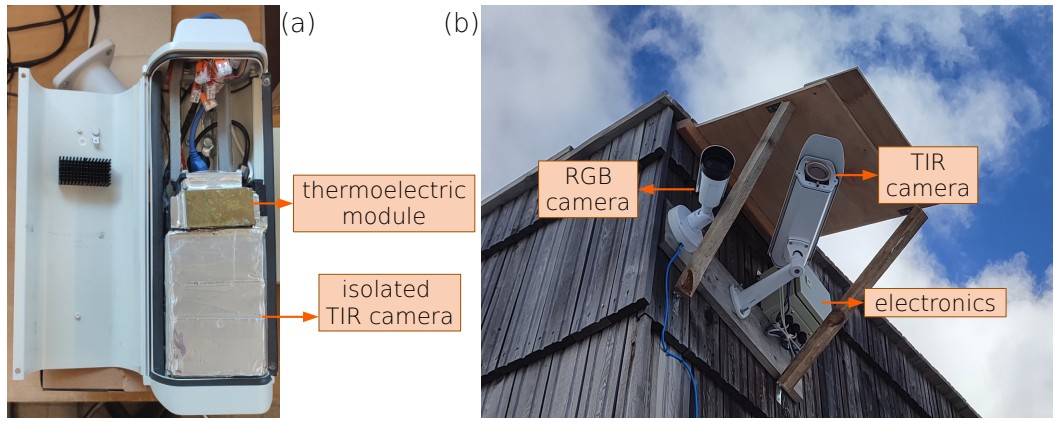

**Figure 3.** The interior of the Optris Pi640 TIR camera's casing in the spring 2023 setup. The camera is isolated in the volume below, the TEM and radiators for heat evacuation are on top (a). The TIR camera on the right and a regular RGB camera (Axis P1448-LE) on the left under the protective roof (b).

## 3.2 Other instruments

### 3.2.1 Visible camera

An RGB camera (Axis P1448-LE) was mounted to the side of the TIR camera (Fig. 3b). The fields of view (FOV) of the two cameras correspond approximately (Fig. 2). Visible images were acquired every 10 minutes during operation of the TIR camera and used as a visualization and interpretation support for the lower-resolution TIR images, especially during foggy or snowy conditions.

### 3.2.2 Automatic Weather Stations

During winter 2021-2022, three Campbell Scientific IR120 TIR radiometers were installed in the camera's FOV, two on the South and one on the North facing slope (Fig. 1 and Fig. 2). During spring 2023, the three radiometers used during the previous season were integrated into two more complete Automatic Weather Stations (AWS), one situated on the North facing slope (AWS-North), one on the South facing one (AWS-South, Fig. 1 and Fig. 2). The list of measurements performed and the instruments employed are described in Table 1, while their setup is illustrated in Fig. 4. TIR radiometer measurements are

used as ground truth to validate the TIR camera measurements. During the second campaign, two high precision Heitronics CT15.85 radiometers were added to the three Campbell Scientific IR120 used during the first campaign after noticing a warm error of the IR120 sensor up to 1.7 K over melting snow. Indeed, these sensors proved to be very precise in controlled laboratory 140 conditions, but are sensitive to changes to the sensor temperature that occur in the field, resulting in a degraded accuracy. The Heitronics radiometers compensate for this effect with a re-calibration that is performed every 10 s approximately.

| Measurement | Model | Description | Installation |
|---|---|---|---|
| Snow surface temperature | Heitronics CT15.85 | TIR radiometer - pyroelectric chopped radiation method, spectral response in the range 9.6 $\mu$m to 11.5 $\mu$m, declared measurement uncertainty $\pm 0.5$ K+0.7$\Delta$T | 1 at AWS-North, 1 at AWS South |
| | Campbell Scientific IR120 | TIR radiometer, spectral response in the range 8 $\mu$m to 14 $\mu$m, declared measurement uncertainty $\pm 0.2$ K | 1 at AWS-North, 2 at AWS South |
| Incoming longwave radiation | Hukseflux IR02 | Measures radiation in the 4.5 to 42 $\mu$m range, the FOV angle is 150° | 1 at AWS-North |
| | Kipp & Zonen CGR3 | Measures radiation in the 4.5 to 42 $\mu$m range, the FOV angle is 150° | 1 at AWS-South |
| Air temperature and humidity | Vaisala HMP155A | Measurement of temperature in the -80°C to +60°C range, relative humidity in the 0 to 100% range | 1 at AWS-North, 1 at AWS South |
| Wind speed | MetOne 014A | Three-cup anemometer, measures wind speed in the 0.45 ms$^{-1}$ to 45 ms$^{-1}$ range with a 1.5% accuracy | 1 at AWS-South |

**Table 1.** List of measurements performed and basic specifications of the instruments installed at the AWS-North and AWS-South at the Col du Lautaret site.

### 3.3 Other measurements

Ground Control Points (GCP), i.e. points of well known location in both the TIR camera and world coordinates, were acquired regularly throughout the two seasons. They were obtained as follows: a 100x70 cm aluminum plate was displaced through 145 the camera's FOV. At specific locations, the plate was laid on the ground and a RTK-GPS measurement of its position with a 0.01 m accuracy was acquired. The aluminum plate has a low emissivity and only reflects the sky temperature which is, in clear sky conditions, much colder than the snow surface. The position of the aluminum plate during the GPS survey was therefore clearly distinguishable in the TIR images. 21 GCPs were acquired during winter 2021-2022, 25 during spring 2023. Their positions both in world and camera coordinates are listed in Table A1 and Table A2 and shown in Fig. A1 and Fig. A2 in 150 Appendix A.

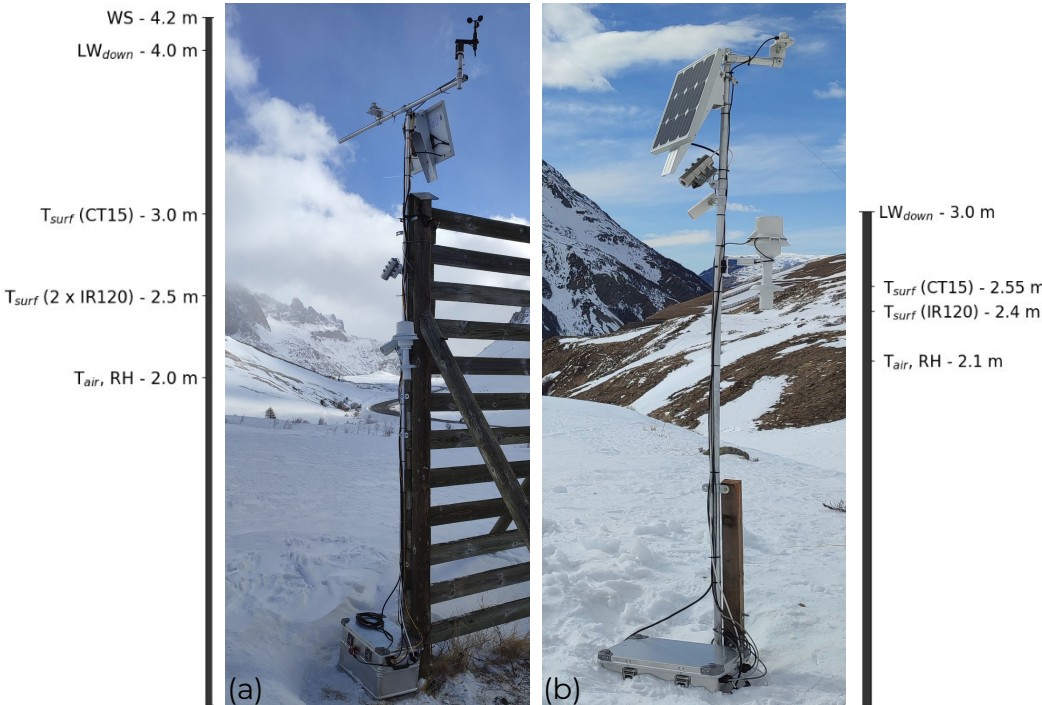

**Figure 4.** Picture and scheme of the instrument setup of the AWS-South (a) and AWS-North (b).

Also, the internal temperature of the TIR camera, $T_{int}$, was measured continuously and recorded every 2 seconds through the COM-port communication tool of the Optris PIX-Connect software (Opt, 2018).

## 4 TIR camera characterization

Between winter 2021-2022 and spring 2023 and after the instrument setup modification, some physical parameters of the TIR camera were measured during laboratory experiments, and its calibration was verified. The results of this characterization were used later to process the data acquired during the two field campaigns.

### 4.1 Assessment of the distortion parameters

An assessment of the distortion parameters of the TIR camera is performed using the *calibrateCamera* function of the OpenCV Python library (Itseez, 2015). More specifically, 55 TIR images were acquired in the lab holding a chessboard in different directions and orientations. The chessboard developed for this application alternates wood (high emissivity) and aluminum (low emissivity) squares in order to enhance the contrast in front of the thermal sensor. During the acquisition, the internal temperature of the camera was set to +15°C. The positions in the images of the chessboard corners were recognized by the *findChessboardCorners* OpenCV function and their position was passed down to the *calibrateCamera* function that computes

the distortion parameters. The resulting radial distortion coefficients $k_n$ and tangential distortion coefficients $p_n$ are:

$$k_1 = -0.42230,\ k_2 = 0.52240,\ k_3 = -0.64157,\ p_1 = 0.00003,\ p_2 = 0.00185 \tag{1}$$

## 4.2 Determination of the camera window model

The camera was located inside an aluminum casing equipped with a Germanium window coated with an anti-reflection film. Still, its transmissivity is slightly below 1, which means that part of the signal measured by the TIR camera's sensor does not come from the target but is either one of 1) the thermal emission of the window towards the sensor, 2) the reflection of the casing's thermal emission onto the window, or 3) the combined effect of 1) and 2). The temperature of both the casing and the window being significantly higher than the snow surface during use, even a small percentage of their emission can increase the measured temperature by many degrees. The manufacturer supplied a transmittivity value of 0.92 but no reflectivity or emissivity values. Moreover, we found this transmittivity to be lower than most Germanium windows with anti-reflection coating (0.95-0.99) available in commerce. Also, atmospheric UV radiation can damage the anti-reflection coating in time and cause these parameters to change. Transmissivity, reflectivity and emissivity ($t$, $r$ and $e$) of the camera's window were thus estimated experimentally to provide a correction.

With this aim, the window was removed from the camera's enclosure and its optical properties were tested in a climate chamber, with the use of a Land P80P black-body source (emissivity $e > 0.995$, uncertainty $|\sigma| < 0.1$ K) and two TIR radiometers, one Heitronics CT15.85 and one Heitronics KT19.85. More specifically, the temperature inside the climate chamber was set to constant 10°C for the window to thermalize at a known temperature. First, the CT15.85 radiometer was placed in front of the black body source and the target temperature was measured at the frequency of 1 Hz for 5 minutes. Second, the measurement was repeated by interposing the thermalized camera window between the radiometer and the black-body target. Third, the KT19.85 radiometer was placed in front of the CT15.85 radiometer in order to measure its brightness temperature. The radiative transfer parameters $t$, $r$ and $e$ are then computed assuming:

$$T_{\text{meas}}^4 = t \cdot T_{\text{targ}}^4 + r \cdot T_{\text{B-rad}}^4 + e \cdot T_{\text{win}}^4 \tag{2}$$

where $T_{\text{targ}}$ and $T_{\text{meas}}$ are the target temperatures measured during step 1) and 2) respectively. $T_{\text{B-rad}}$ is the CT15.85 radiometer's brightness temperature measured during step 3) and $T_{\text{win}}$ is the camera window temperature, assumed to be equal to the ambient temperature inside the climate chamber. To retrieve the three parameters, the whole experiment was repeated for 3 different black-body temperatures (20°C, 12.5°C, 5°C). Considering Kirchoff's law of thermal radiation $t + r + e = 1$, the result is:

$$t = 0.95,\ r = 0.03,\ e = 0.02 \tag{3}$$

## 4.3 Calibration assessment

The camera is, in principle, calibrated by the manufacturer. To test this calibration in the typical conditions for our use, the camera was put in a cold chamber using the black-body source as target of well-known temperature. Tests were run with an ambient temperature of 10°C, 0°C and -10°C and black-body temperatures of 5°C, 0°C, -5°C and -10°C. All combinations of

ambient and black-body temperatures above were tested with the camera internal temperature set to 15°C and 22°C. For each of the resulting 24 temperature combinations, 500 images were acquired at the frequency of 1 Hz. During this procedure, the FFC was performed every 12 s, as in field conditions. The Landcal P80P black-body source has a diameter 50 mm and is located at the end of a 50 mm wide and 160 mm long cavity. It was therefore not possible to approach the TIR camera close enough to the black body source to match the field of view and the area of homogeneous temperature. The calibration assessment was thus carried out using the central 240x210 pixel rectangle shown in Fig. 5. The analysis of the images raised two issues, 1) two areas of defective pixels, and 2) a dependence of the raw measurements on the difference between the camera's internal temperature and the TEM setting temperature $T_{int}$-$T_{set}$, not explained by the camera window's emissive contribution.

### 4.3.1 Defective pixels

Two areas with degraded pixels were identified. In the upper part of the image (white rectangle in Fig. 5) some pixels were damaged, likely because of the impact of direct sunlight onto the sensor during the first year of installation. This defect was ignored as these pixels contained parts of the sky or mountains in the far horizon that were not useful for the scope of the dataset. In the left part of the image (orange rectangle in Fig. 5) a circular shape was subject to a warm bias. Its cause is unknown. For each image of the calibration assessment, the warm error was computed by subtracting the average black-body temperature measured excluding the warm area. Then, the average warm error of all images in the selected area was computed. Its magnitude varied spatially, as the bias is stronger at the center of the warm area, up to 0.54 K, and decreased towards the margins (see Fig. 5). The magnitude of the warm error was stable over time for each pixel, with a standard deviation that varies spatially but it's $\leq 0.22$ K everywhere. The magnitude of the warm error was therefore considered constant from image to image for every pixel in the warm area.

### 4.3.2 Dependence on $T_{int}$-$T_{set}$

The difference between the averages of the temperatures measured for each internal and target temperature combination $T_{meas}$ and the corresponding black-body temperatures $T_{bb}$ are represented in Fig. 7a and Fig. 7b for a set internal temperature $T_{set}$ of 15°C and 22°C respectively. The raw measurements and the black-body temperatures corrected as if they are observed through the window $T_{bb\text{-}win}$ should be equal if the camera is perfectly calibrated. This is not the case, instead we found that the difference between these two temperatures was proportional to the difference between the TEM setting temperature $T_{set}$ and the real camera internal temperature $T_{int}$ (see Fig. 7c). Through a linear regression, the following relation was found:

$$T_{meas} - T_{bb\text{-}win} = 1.92 \cdot (T_{int} - T_{set}) - 1.33 \tag{4}$$

## 5 Processing

The processing of the TIR images consists in three main steps. First, the radiometric processing improves the quality of the measurements by correcting the instrumental issues described in section 4. Second, images with poor or null visibility of

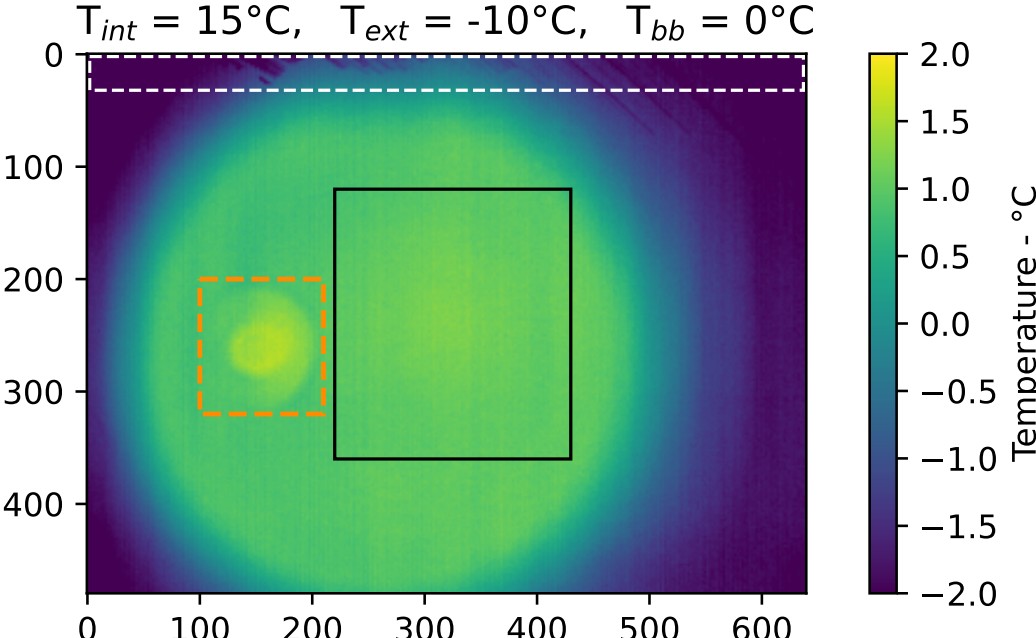

**Figure 5.** Image of the black body target (0°C) highlighting two areas with defective pixels. In the upper part, pixels damaged by direct sunlight exposure during the first year of deployment (white rectangle). On the left, pixels subject to a warm error of unknown cause (orange rectangle). On the right, the area used for calibration (black rectangle).

the surface were deleted from the dataset. Third, during the geometric processing, image distortions were corrected, slight movements of the camera observed throughout the season were compensated, and the images were orthorectified, i.e. projected onto a digital elevation model (DEM). The whole processing is summarized in the workflow in Fig. 6. Our method does not include corrections for non-instrumental effects such as emissivity and atmospheric contributions. We therefore consider these effects to be included in our error estimations.

## 5.1 Radiometric processing

The radiometric processing was performed in three steps. First, the warm error described in Paragraph 4.3.1 was corrected. As the bias is considered to be constant in time, the average image of the warm error acquired during the TIR camera's characterization was subtracted from every image. Second, according to equation 4, the bias due to the $T_{\text{int}}$-$T_{\text{set}}$ dependency was corrected as follows:

$$T_{\text{meas\_corr}} = T_{\text{meas}} - 1.92 \cdot (T_{\text{int}} - T_{\text{set}}) + 1.33 \tag{5}$$

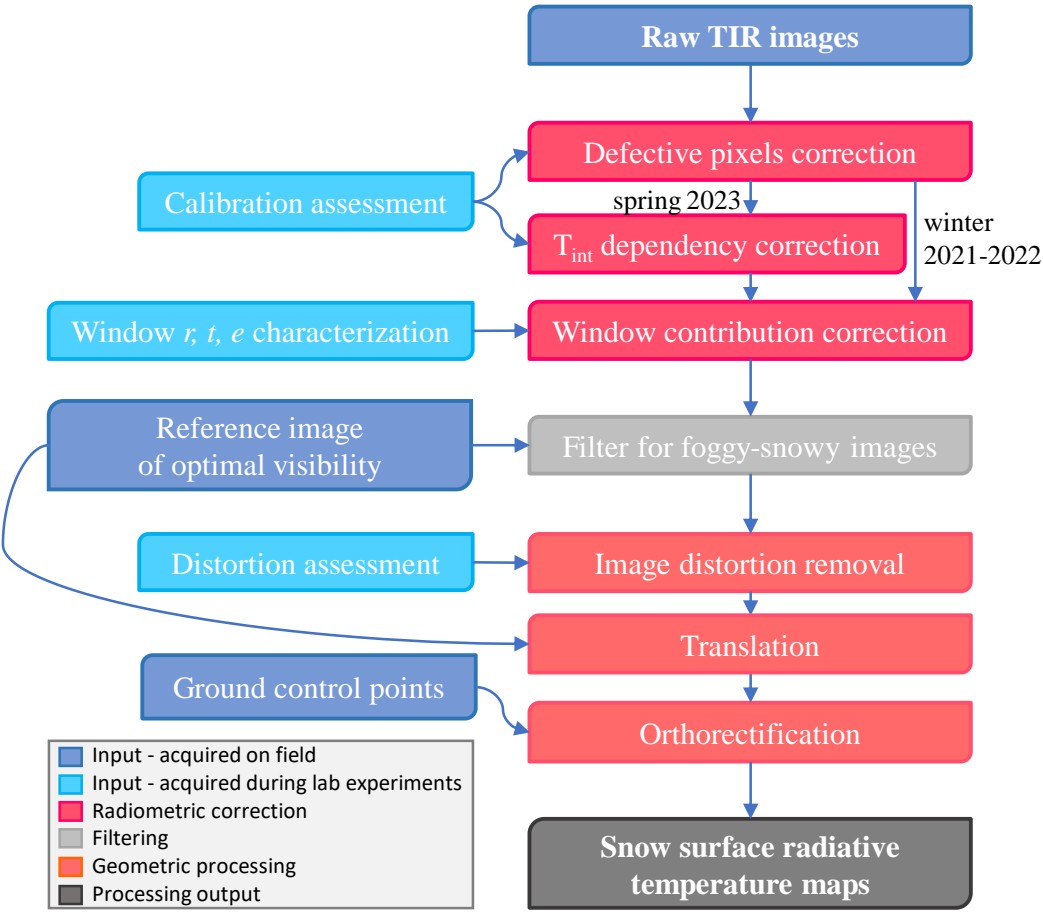

**Figure 6.** Workflow of the processing of raw TIR images to radiometrically-corrected and orthorectified surface radiative temperature maps.

Third, the effect of the camera's window was corrected:

$$T_{\text{meas\_rad\_corr}} = \left( \frac{(T_{\text{meas\_corr}} + 273.15)^4 - r \cdot (T_{\text{int}} + 273.15)^4 - e \cdot (T_{\text{win}} + 273.15)^4}{t} \right)^{0.25} - 273.15 \qquad (6)$$

where $T_{\text{win}}$ is taken as the average of $T_{\text{int}}$ and $T_{\text{ext}}$. $T_{\text{ext}}$ is measured at the AWS-South station, the closest to the camera. If not available, $T_{\text{ext}}$ was taken from the automatic weather station FluxAlp (45.0413 N, 6.4106 E, Laurent et al., 2012) located on the East side of the Col du Lautaret, about 1 km from our site. The differences between the corrected measurements and the black-body temperatures are shown in Fig. 7d for an internal temperature of 15°C and Fig. 7e for an internal temperature of 22°C. The residual errors span between -0.57 K and +0.38 K and are distributed around 0 °C with a standard deviation of 0.22 K.

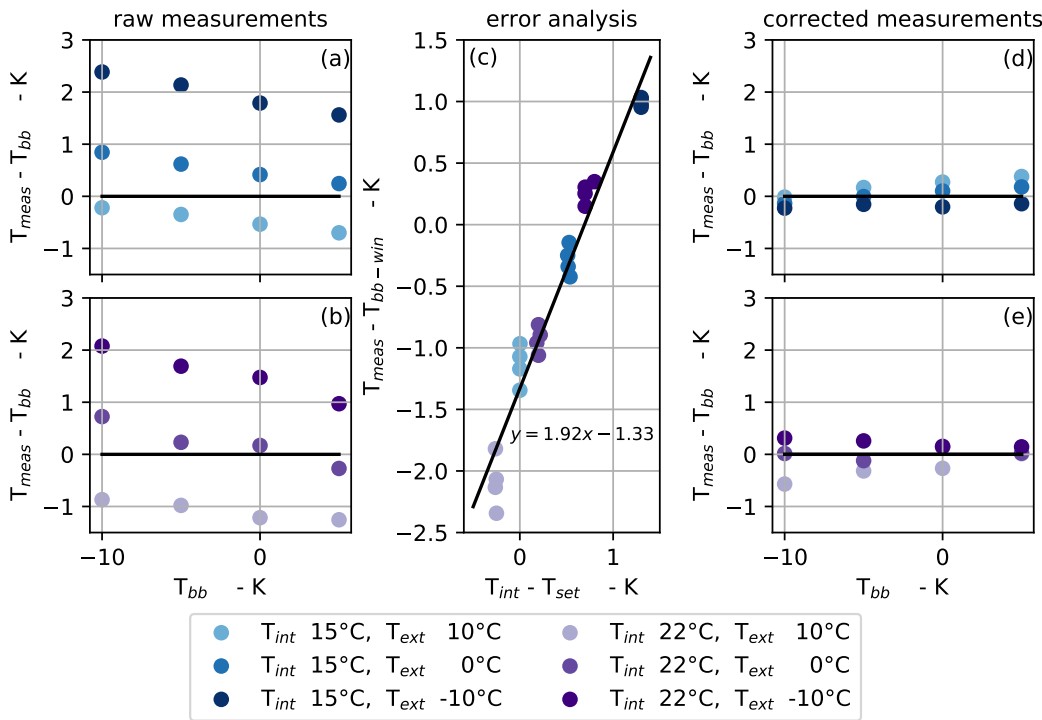

**Figure 7.** Difference between the average measured temperature $T_{meas}$ and the actual black-body temperatures $T_{bb}$ as a function of black-body temperatures, for internal temperatures $T_{int}$ of 15°C (a) and 22°C (b). The difference between the raw measurements and the black-body temperatures corrected as if they are observed through the window $T_{bb\text{-}win}$ as a function of the difference between the thermoelectric module setting $T_{set}$ and the real camera internal temperature $T_{int}$ (c). The differences between the corrected measurements and the black-body temperatures for an internal temperature of 15°C (d) and 22°C (e).

## 5.2 Cloud filtering

Transiting clouds and snowfall reduce the visibility of the surface from the TIR camera. To detect the presence of clouds, we relied on the detection of different objects that are clearly recognizable during days with optimal visibility. With this aim, the *MatchTemplate* function of the OpenCV Python library identifies the position of a given feature (i.e. a small reference image) in a complete TIR camera image as the location where the two overlapping images correlate most. The chosen reference image was taken at solar noon of one sunny day per year (2022-03-05 13:00 UTC+1 and 2023-03-05 13:00 UTC+1). The only

snow-free features in the camera's FOV are the road and the vegetation. Both are warmer than the surrounding snow-covered soil.

Cloud filtering was performed in two steps. First, for every image, a 356x200 cropped image including a portion of the road was extracted (orange rectangle in Fig. 8). *MatchTemplate* computes the position of the cropped image into the reference image. If the returned position had an the absolute error ≥5 pixel for any of the x or y dimensions, the image was considered cloudy

and rejected. If the returned position is correct up to an absolute error <5 pixel for both dimensions, the road was considered recognized. The small detected displacements (0 – 4 pixels) of the evaluated image with respect to the reference image were considered as plausible, since the camera moves slightly throughout the season. Second, two smaller features (trees), on the left and right sides of the image, were cropped from both the reference and the evaluated image (white rectangles in Fig. 8). The Pearson correlations with respect to the reference were computed with *MatchTemplate*. If any of the two correlations

was ≤0.25, the features were considered non-recognized and the whole image was considered cloudy. It was therefore rejected. Otherwise, the image was definitely considered having sufficient visibility.

This method was tested against a set of ≈1000 manually chosen TIR images with good visibility on snow-covered terrain and ≈1000 images with conditions form mist to thick fog. Among the days with clear sky, only 1 image was discarded, which means a 0.1% false positive error. Of the images with poor visibility, 19 were retained, 1.9% false negative. Despite

this encouraging performance, the method proved to be inefficient at the beginning and at the end of the winter season, as the presence of large bare ground patches altered the surface temperature patterns that no longer correlate with the snow-covered ground in the reference image. Thus, the classification for the images acquired before the 25-11-2021 and after the 2022-04-13 for winter 2021-2022, and after the 2023-04-19 for spring 2023, was performed visually.

## 5.3 Geometric processing

The geometric processing aims at obtaining accurate orthorectified maps of the snow surface temperature from radiometrically corrected and cloud-filtered TIR camera acquisitions. It is performed in three steps. First, image distortions were removed using the *undistort* function of the OpenCV Python library and the distortion parameters obtained during the TIR camera characterization. Second, the images were translated in order to compensate for the slight movements (up to 4 pixels) of the TIR camera observed throughout the two seasons. The displacement was computed with the *MatchTemplate* function of the

OpenCV Python library as in Section 5.2. The images were translated by the median displacement of the day of acquisition using the *warpAffine* function of OpenCV.

The orthorectification, which is the projection of the camera images onto a Digital Elevation Model (DEM), was finally performed using the Ames Stereo Pipeline from NASA (ASP, Beyer et al., 2018) in two steps. First, the camera position is computed using the camera's focal length, the DEM and GCPs. For this step, a 50 cm resolution DEM acquired by LIDAR

on bare soil was used. Before the orthoprojection, 0.75 m were added to the DEM height to simulate an average snow cover height for the season. The coordinates (XY) of the GCPs in the TIR camera reference were extracted from the images after the translation. The camera position, that integrates the camera model, was obtained with the *cam_gen* function of ASP. Second, the *mapproject* command was used to project the images onto the DEM using the refined camera model. The output is a map of the surface temperature in the geotiff format (464x506 pixels, 2 m resolution, singleband, float32 data type, LWZ compression),

which is practical for import in most geographic information software.

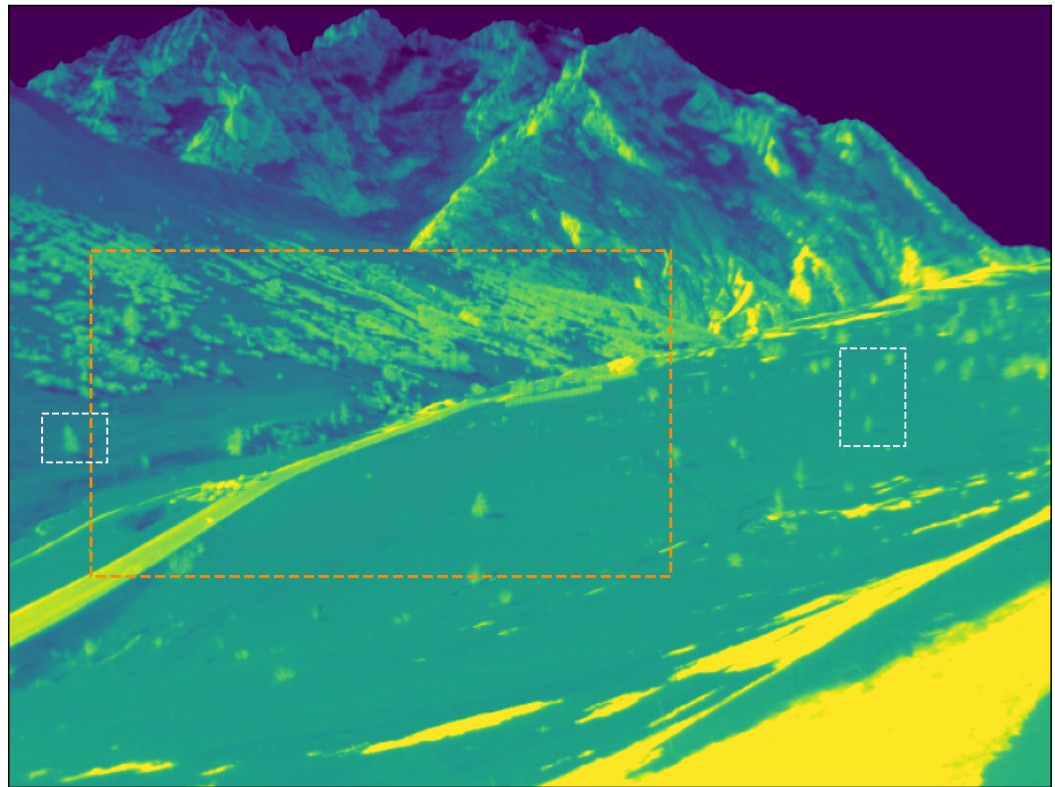

**Figure 8.** Areas of the image used to evaluate the overall visibility of the picture, to detect displacement (orange rectangle) and to evaluate the visibility of small features (two vegetation features, white rectangles).

## 6 Results

The number of acquired TIR images of the Col du Lautaret site is over 103000 for winter 2021-2022 and over 61,000 for spring 2023. The duration of the measurement periods is described in Fig. 9. During winter 2021-2022, the interruptions of the TIR camera measurements are due to electricity outages on site, while the missing TIR radiometer measurements are due to battery problems that were solved during the spring. The missing measurements at the beginning of spring 2023 are due to camera overheating episodes. Partial snow cover is indicated for periods when the snow is not present yet or already melted at AWS-North and AWS-South but the soil at the Col du Lautaret remains partially covered with snow.

The distributions of the raw and radiometrically corrected surface temperature measurements acquired at AWS-South and AWS-North during the two seasons are shown in Fig. 10. The values were extracted from the images before orthorectification, and correspond to the the average of 4 pixel squares corresponding to the radiometer's fields of view. The radiometric correction caused a decrease in the median measured temperature for all distributions. The decrease was more important for the winter 2021-2022 distributions than for the spring 2023 ones, with an average decrease of of 1.62 K against 0.55 K. This difference is due to two factors. First, the stronger correction for the window transmission performed for winter 2021-2022, that causes

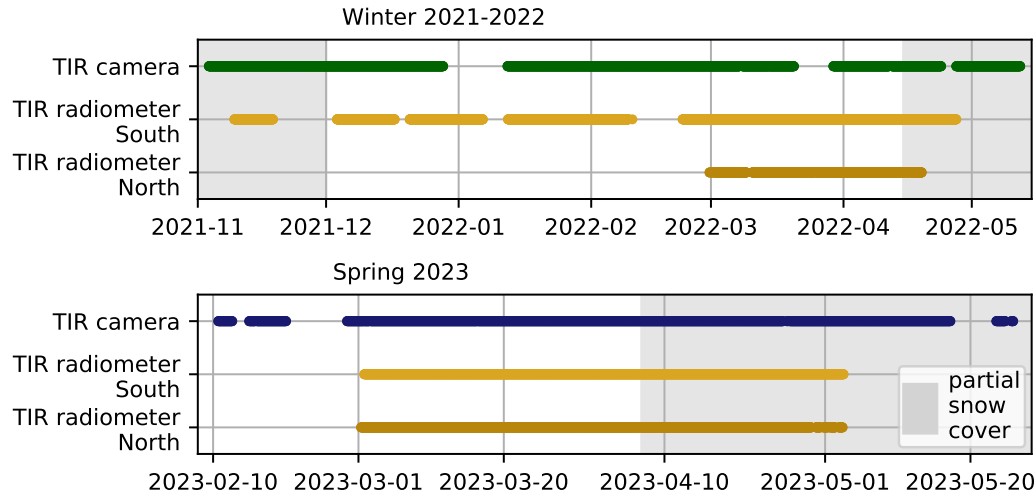

**Figure 9.** Illustrations of the periods of measurement of the surface radiative temperature of snow throughout winter 2021-2022 and spring 2023. Periods of partial snow cover, indicating no snow cover at AWS-South and AWS-North, are shaded.

the whole decrease of the measured temperature of 1.62 K during winter 2021-2022 and a decrease of only 1.12 K during spring 2023. This was caused by the higher average internal temperature of the TIR camera during the first campaign, 21.9°C against 19.3°C for spring 2023. Second, the correction for the dependence on $T_{int}$-$T_{set}$ performed for measurements of spring 2023 causes an average increase of 0.66 K everywhere on the image. Finally, the measurements acquired at AWS-North during spring 2023 are the only ones affected by the warm pixel area described in section 4.3.1 among the presented distributions. The correction for the warm pixel areas causes a decrease of 0.17 K for all images.

The filtering of the images of unsatisfactory visibility resulted in 77% of the images from winter 2021-2022 and 82% of the images from spring 2023 being retained. An additional 16,600 measurements from the first season and 211 from the second one were excluded from the dataset as no measurement of the camera's internal temperature was available to apply the radiometric processing. As a result, a total of 130,019 images (79,563 from 2021-2022 and 50,456 from 2023) went through the geometric processing.

The triangulation of the GCPs on the DEM determines similar camera positions for the two seasons, illustrated in Table 2. Both computed positions were higher than the real one by 13/12 m and are 33/39 m NE of the real camera position. The median projection error of the GCPs was 4.3 m for winter 2021-2022 and 3.2 m for spring 2023, which was considered sufficient for the comparison with multi-decameter resolution satellite images. The projection error for each GCP is listed in Table A1 and Table A2 in Appendix A.

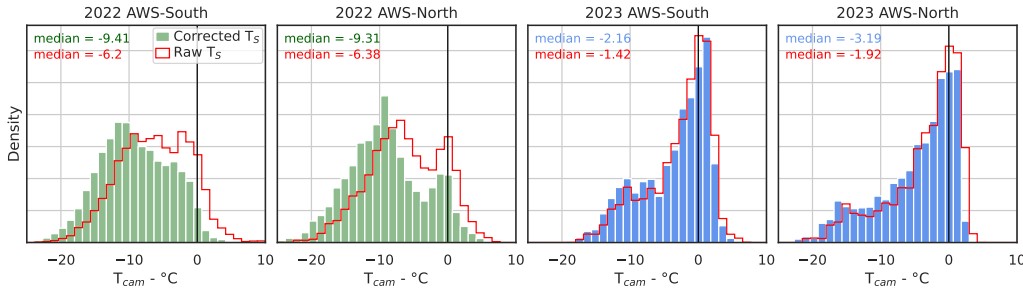

**Figure 10.** Distributions of the measured raw (red) and radiometrically corrected (green and blue) surface radiative temperatures of snow at AWS-South and AWS-North throughout winter 2021-2022 and spring 2023.

| Dataset | Longitude | Latitude | Altitude |
|---|---|---|---|
| Winter 2021-2022 | 6.401023° | 45.035436° | 2159 m |
| Spring 2023 | 6.400938° | 45.035375° | 2158 m |
| Real position | 6.400700° | 45.035200° | 2146 m |

**Table 2.** Camera positions computed by the Ames Stereo Pipeline *cam_gen* function for the two datasets from the camera's focal length, the DEM and the GCPs, compared to the real camera position.

## 7    Validation

The validation of the two-year dataset was performed using TIR radiometer data acquired at AWS-North and AWS-South, in the TIR camera's FOV. The radiometers' specifications are described in Table 1 and their position is shown in Fig. 1 and Fig. 2. The performance of the surface temperature measurement $T_{cam}$ with respect to the ground truth $T_{rad}$ is shown in table 3. The error distributions are described using the median and the Mean Absolute Error (MAE) in order to consider both the precision and the accuracy of the measurements. All distributions having median or MAE ≤0.7 K (marked in bold) correspond to measurements acquired during spring 2023. Indeed, the lowest surface temperatures ($\leq -1°C$) measured during the second season by the TIR camera have a slightly negative to positive median bias with respect to the TIR radiometer measurement, between -0.02 K and +0.49 K. The MAEs meet the <0.7 K target except for the temperatures -5°C≤$T_{cam}$≤-1°C measured at AWS-South and the temperatures ≤-5°C measured at AWS-North, which are slightly above (0.71 K and 0.83 K respectively). The larger MAE at AWS-North in particular may be due to additional noise induced by the warm defective pixels affecting the area. The highest measured snow temperatures (>-1°C) during 2023 have the highest median difference with respect to $T_{rad}$, +1.17 K at AWS-South and +0.61 K at AWS-North. Indeed, during spring 2023, the measured surface radiative temperature measured by the TIR camera over melting snow is found to be higher than freezing point (0°C) by several degrees, as shown in Fig. 11 (red circle). The cause of this temperature overshoot is unknown. The median and the MAEs are therefore much lower if only temperature $< -1°C$ are considered (0.22 K and 0.68 K respectively) compared to for the whole distribution (0.48 K

and 0.82 K respectively). The errors of the measured surface temperatures from winter 2021-2022 are always negative and range between -0.74 K and -1.95 K, with a median error of -1.05 K, and a MAE of 1.28 K. None of the medians or MAEs for the first season meets the 0.7 K target. Some overshoots of the surface radiative temperature at 0°C are also observed during winter 2021-2022. However, they are less visible in Fig. 11 because of the cold bias of the overall distribution and of the reduced precision of the IR120 radiometers. Also illustrated in Fig. 11 are the linear regressions of the two distributions (orange lines), the computed slopes, intercepts, fit $R^2$ and RMSE (respectively $m$, $b$, $R^2$ and $\sigma_y$). Both distributions confidently fit to the regression, as shown by the elevated $R^2$ -0.96 for winter 2021-2022, 0.97 for spring 2023-. The first has an intercept $b$=-1.81°C with the error decreasing with the temperature as the slope $m$=0.91. The second, on the other hand, has an intercept $b$=+0.52°C and the bias is fairly constant as $m$=1.02.

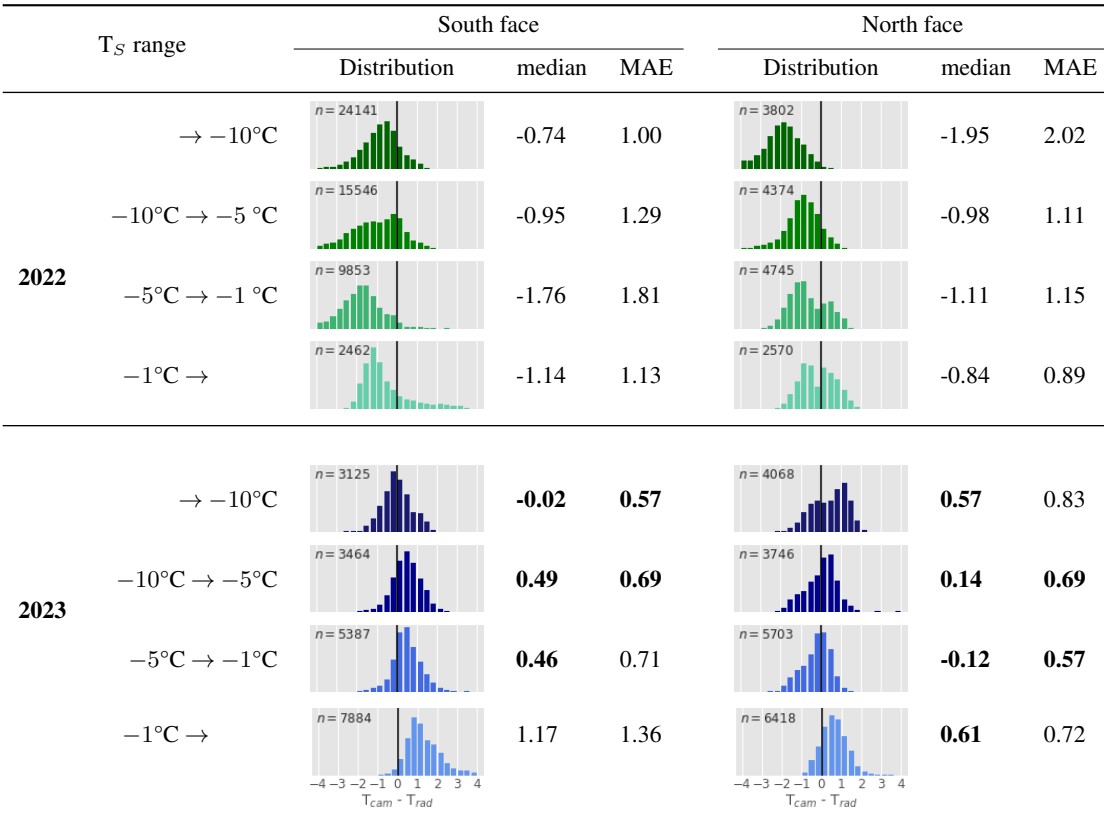

| $T_S$ range | South face | | | North face | | |
|---|---|---|---|---|---|---|
| | Distribution | median | MAE | Distribution | median | MAE |
| **2022** | | | | | | |
| $\rightarrow -10$°C | $n=24141$ | -0.74 | 1.00 | $n=3802$ | -1.95 | 2.02 |
| $-10$°C $\rightarrow -5$ °C | $n=15546$ | -0.95 | 1.29 | $n=4374$ | -0.98 | 1.11 |
| $-5$°C $\rightarrow -1$ °C | $n=9853$ | -1.76 | 1.81 | $n=4745$ | -1.11 | 1.15 |
| $-1$°C $\rightarrow$ | $n=2462$ | -1.14 | 1.13 | $n=2570$ | -0.84 | 0.89 |
| **2023** | | | | | | |
| $\rightarrow -10$°C | $n=3125$ | **-0.02** | **0.57** | $n=4068$ | **0.57** | 0.83 |
| $-10$°C $\rightarrow -5$°C | $n=3464$ | **0.49** | **0.69** | $n=3746$ | **0.14** | **0.69** |
| $-5$°C $\rightarrow -1$°C | $n=5387$ | **0.46** | 0.71 | $n=5703$ | **-0.12** | **0.57** |
| $-1$°C $\rightarrow$ | $n=7884$ | 1.17 | 1.36 | $n=6418$ | **0.61** | 0.72 |

**Table 3.** Distributions and statistics of the difference of surface temperature measured by the camera $T_{cam}$ and the radiometers $T_{rad}$. Distributions meeting the target of median and MAE $\leq$ 0.7 K are marked in bold.

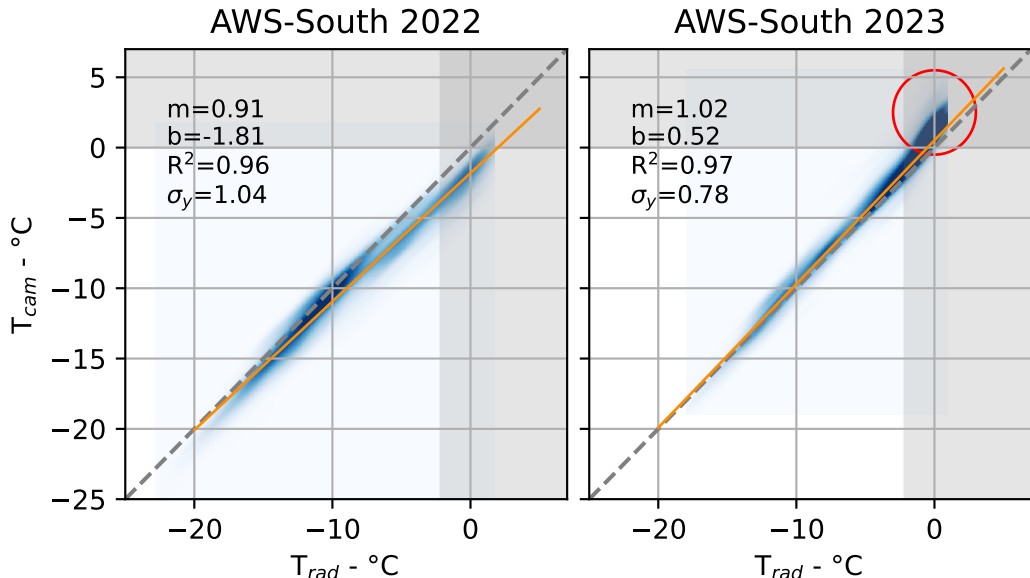

**Figure 11.** Density distributions of the snow surface temperatures measured at AWS-South by the camera as a function of those measured by the radiometer during winter 2021-2022 (a), and spring 2023 (b). The plot shows the high measurements of the surface temperature measured during spring 2023 over melting snow (red circle). Also shown are the identity line (dotted gray) and the linear regression of the distributions (orange).

## 8 Discussion

Surface radiative temperature maps of the snow cover were acquired at the Col du Lautaret over winter 2021-2022 and spring 2023 with an uncooled TIR camera. The maps were continuously validated against radiative surface temperature measurements performed by two high precision TIR radiometers in the camera's field of view. This dataset also includes measurements of the incoming longwave radiation, air temperature and humidity, wind speed measured on site, the TIR radiometer temperature measurements, internal TIR camera temperature timeseries and RGB images of the site.

The processed measurements acquired during winter 2021-2022 exhibit an median error of -1.05 K and a MAE of 1.28 K, both above the specified accuracy target of 0.7 K. The lower accuracy is clearly attributed to the inherent instability of the internal temperature of the Thermal Infrared (TIR) camera throughout the season. Indeed, the camera operated, in its commercial configuration, resulting in a broad range of internal temperatures (7°C to 39°C) whose impact is not correctly compensated by the internal FFC. Due to the absence of an internal temperature reference, we are not able to apply retroactively the corrective measure for the variations in the internal temperature applied for spring 2023 to the 2022 dataset. Nevertheless, the 2022 dataset remains useful, as with an accuracy of 1.28 K, our measurements perform better than the 1.5 K (1-$\sigma$) target for the level 2A LST product of TRISHNA. To facilitate further investigation, the processed maps can be adjusted for the surface temperatures

to match the ground truth (the average TIR radiometer measurement recorded at AWS-North and AWS-South), when available. The accuracy of the adjusted measurements is expected to be better.

During the second season, the instrumental setup was modified to limit the variations of the internal temperature of the camera, resulting in a clear improvement of the measurement's quality. With the internal camera temperature constrained, the observations exhibited both improved precision and accuracy with respect to the 2021-2022 winter season, achieving the targeted errors. Overall, the median TIR camera measurement error relative to the ground truth (TIR radiometers, $T_{cam}$-$T_{rad}$) was reduced by a significant 54% with respect to the previous season, going from -1.05K to +0.48 K. More impressively, the median error considering only measurements performed on cold snow ($T_{cam}$<-1°C) was as low as +0.22 K, overcoming the absolute temperature issues encountered by numerous previous studies (Aubry-Wake et al., 2015; Kraaijenbrink et al., 2018; Pestana et al., 2019). The MAE was equal to 0.82 K on the whole season and 0.68 K when considering the measured temperatures $< -1$°C only. We tested the possibility that the higher error for melting snow might be due to a thin layer of warmer water above the snow grains. Still, assuming a thermal conductivity of water $k$=0.6 and a water film thickness of 0.5 mm, 1200 $Wm^{-2}$ of absorbed energy would be needed to sustain a 1 K difference between the top layer and the bottom snow layer at 0°C, which is not plausible. Therefore, the physical cause of this error is unidentified and remains to investigate. However, this error is always a warm bias and therefore easy to identify and correct, as the snow temperature is physically limited at 0°C when the pixels are fully covered by snow. Such correction can be applied to periods when snow on the ground at the two AWS is clearly recognizable on the images acquired by the RGB camera throughout the season.

Another limitation of the current dataset concerns the emissivity. We assume snow is a perfect black-body and do not attempt to apply an emissivity model due to the lack of consensus on the topic. However, snow emissivity is known to be slightly lower than 1 (Hori et al., 2014) and to decrease with the observation angle, especially for angles over 45° in the 12-14 $\mu$m TIR window (Dozier and Warren, 1982; Hori et al., 2006). We assume this error is included in the global error, assessed by the comparison with the ground truth. To facilitate further analysis by the data users, a map of the observation angles of the area, computed as in Castel et al. (2001), is included in the dataset. Moreover, downwelling longwave measurements performed at the Col du Lautaret site are supplied (FluxAlp measurements for winter 2021-2022, AWS-North and AWS-South measurements for spring 2023). With these measurements, the surface radiative temperatures can be adjusted to accommodate emissivity values $< 1$.

Similarly, the measured temperature depends on the radiative contribution of the air between the sensor and the target, that was not corrected and is therefore considered part of the error measured in the comparison with the ground truth at the two AWS. Still, as this error increases with the viewing distance, a distance-to-camera map is included in the dataset, and can be used with temperature and humidity measurements from the AWS to correct for the atmospheric contribution.

Dataset users should account that the camera FOV also contains not only snow but also rocks, trees, a road and wood fences. Moreover, during the orthorectification process, some vertical features such as tall trees or wood fences are projected over long distances on the DEM behind their actual position because of the oblique angle at which the camera is installed. This is always the case behind the snow fences in the central part of the image and can happen when cars are driving on the road in the camera FOV (Fig. 2).

Overall, obtaining accurate measurements of the surface temperature with a TIR camera is challenging and time consum-
ing, as it requires a large number of complementary measurements and laboratory experiments (see Fig. 6). The radiometric
correction compensates a combination of warm bias (window contribution, warm pixels) and cold bias ($T_{int}$-$T_{set}$ dependence).
Thus, for accurate measurements, it is fundamental to compute all bias correctly in order to avoid an apparent accuracy due to
the compensation of positive and negative errors. As such, we recommend to perform a comprehensive characterization of the
camera and test the calibration frequently, at least once per winter season.

All this considered, we conclude that the efforts put in to stabilize the internal temperature of an Optris Pi640 TIR camera
led to accurate measurements of snow surface radiative temperature, achieving an overall accuracy below 0.7 K, which is
comparable to the radiometric accuracy expected for the TIR instrument of the TRISHNA satellite and the high resolution TIR
sensors that will follow.

Examples of timeseries in Fig. 12 illustrate the potential of the dataset to characterize the drivers of snow surface temperature.
Two timeseries (a and b) were acquired during sunny, clear-sky periods. Timeseries (a) was acquired in spring and show similar
daily surface temperature cycles for AWS-South and AWS-North. Indeed, the two areas in spring have comparable exposure
to sunlight, that drives the surface temperature causing the cosine shape of the profile in the middle of the day. Likewise, in
timeseries (b) acquired during a cloudy period, the temperature profiles of the two sites are very similar, showing that the energy
budget of the surface is driven by terms that are barely affected by the topography. In timeseries (c), however, the temperatures
differ significantly. The cosine shape is significantly smaller at AWS-South than in timeseries (a) and absent at AWS-North, as
during winter AWS-South receives some sun while AWS-North remains in the shadow. This illustrates how different processes
can drive the surface temperature within a distance as small as 300 m between the two sites (i.e. less than MODIS, VIIRS or
Sentinel 3 SLSTR pixel size) and at similar altitude. The full timeseries are shown in Fig. 13. Finally, the images in Fig. 14,
acquired under different atmospheric conditions, show the variability of the patterns of the surface temperature of snow caught
by the TIR camera. Beside these examples, the dataset provides numerous other situations that deserve further exploration.

## 9    Conclusions and perspectives

This study presents a dataset of 130,019 georeferenced geotiff images of the surface radiative temperature acquired at the Col
du Lautaret, in the French Alps. Thermal infrared images were acquired with an Optris PI640 uncooled TIR camera during
winter 2021-2022 and spring 2023. The resulting radiative surface temperature maps from the first season have a radiometric
accuracy of 1.28 K (mean absolute error) after processing. The TIR camera was run with an improved setup during spring
2023, reaching a radiometric accuracy of 0.67 K (mean absolute error) for surface temperatures below -1°C and 1.17 K for
melting snow. The results show how the internal temperature stabilization of the TIR camera integrated in the spring 2023
setup strongly benefit both the precision and the accuracy of the measurements, overcoming the original issues. This study
also confirms that performing accurate measurements of the surface temperature of snow with TIR cameras is challenging and
requires camera adaptation, additional field measurements, time-consuming laboratory experiments, and in any case despite
the level of effort, high precision TIR radiometers as ground-truth to achieve the absolute accuracy target. Still, the obtained

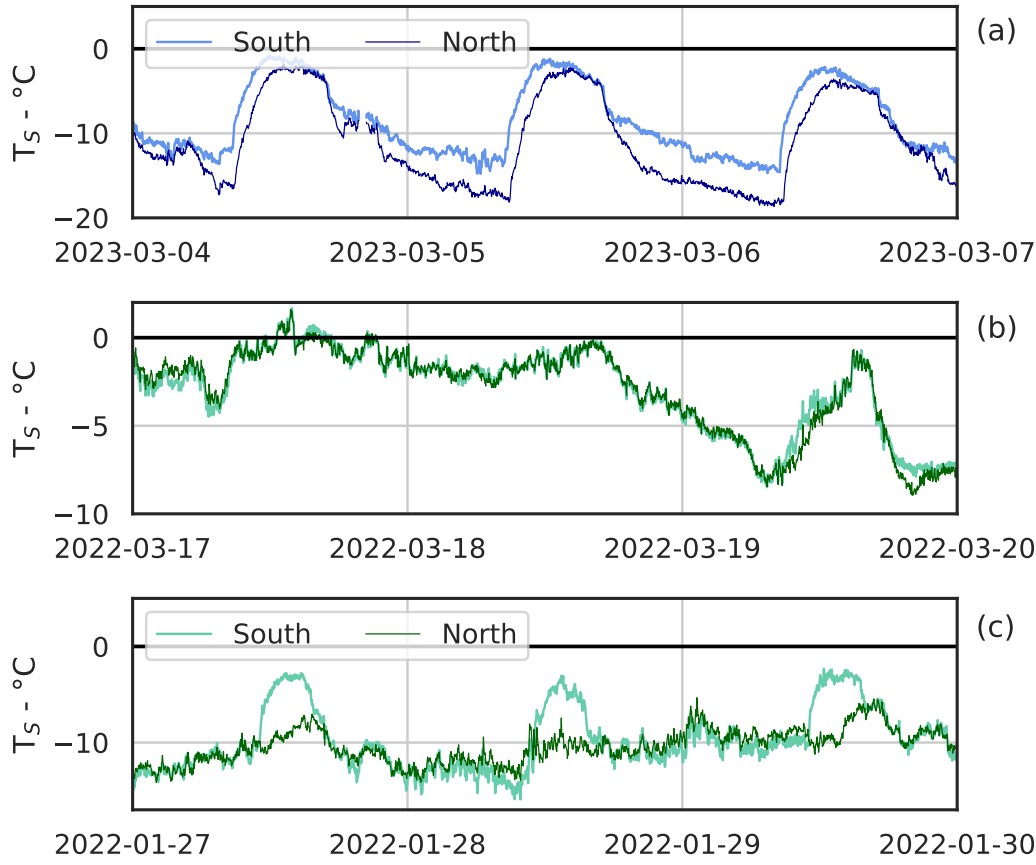

**Figure 12.** Three examples of the surface temperature timeseries measured by the TIR camera at AWS-South and AWS-North during a sunny, spring period (a), a cloudy period (b) and a sunny winter period (c).

measurement accuracy is comparable to that of multiple space-borne TIR sensors and better than most derived LST products. For this, this dataset is a unique to our knowledge and should benefit the investigation of topographic effects on remote sensing and the link between snow emissivity, snow grain characteristics and observational angles. Also, the measurement technique

described in this study represents a potential advance for the calibration-validation of satellite-derived surface temperature products in regions characterized by snow cover and complex topography, providing an evaluation tool for thermal infrared products from Landsat or ECOSTRESS. The know-how gained from this study will be instructive for the cal/val of upcoming, high-resolution thermal infrared missions, starting with TRISHNA at the end of 2026. Finally, considering the increasing use of Unmanned Aerial Vehicle (UAV) measurements employing Thermal Infrared (TIR) cameras over cryospheric surfaces

(Gök et al., 2023; Rossini et al., 2023; Wigmore and Molotch, 2023), the current configuration for a stationary camera serves as a foundational framework for future UAV applications. In fact, the first trials of UAV measurement of the snow surface temperature with a TIR camera with a similar improved setup are underway.

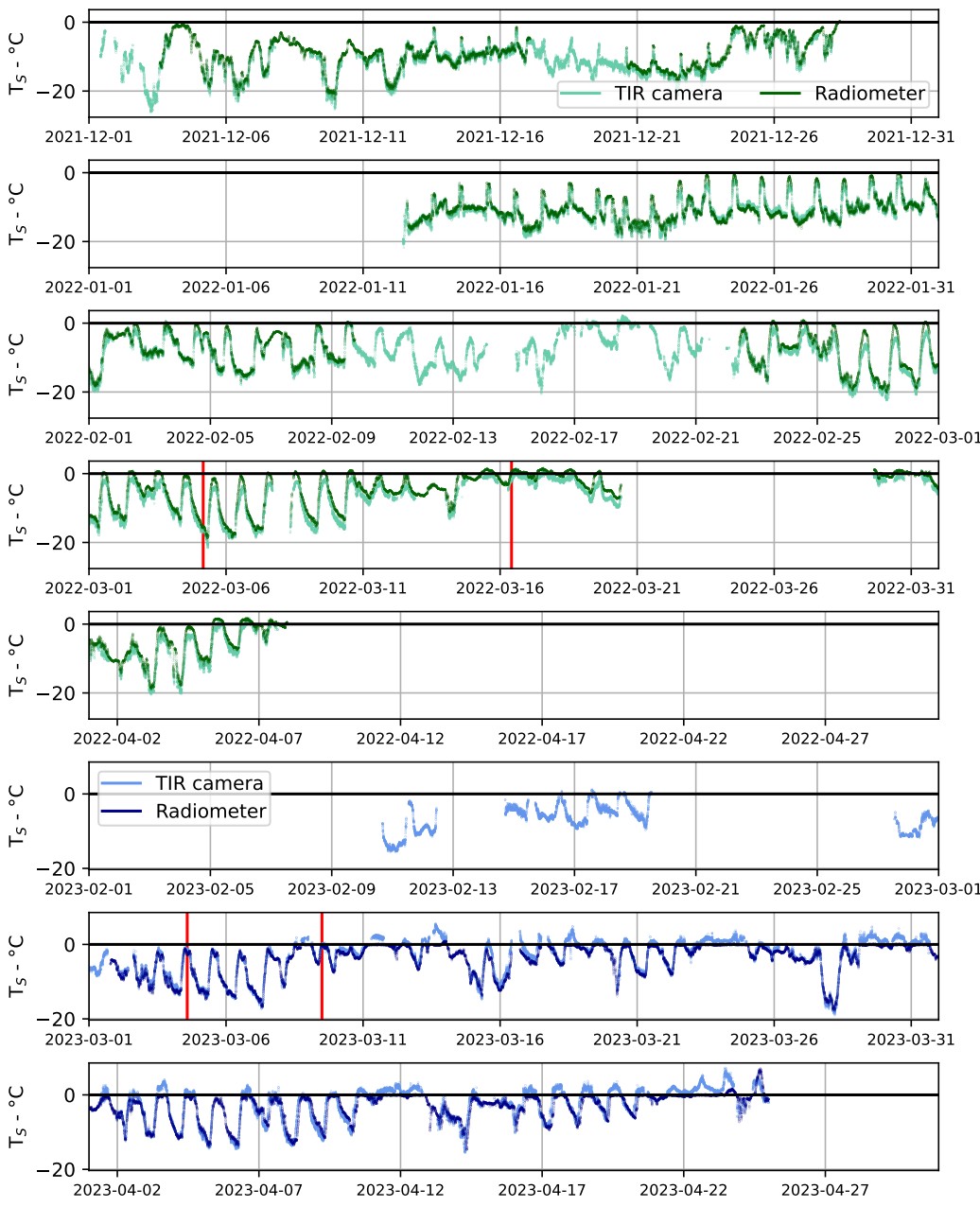

**Figure 13.** Full timeseries of the snow surface temperature measured by the TIR camera and by the CT15.85 radiometer at AWS-South over the two seasons, winter 2021-2022 in green and spring 2023 in blue. The red lines correspond to the images of Fig. 14.

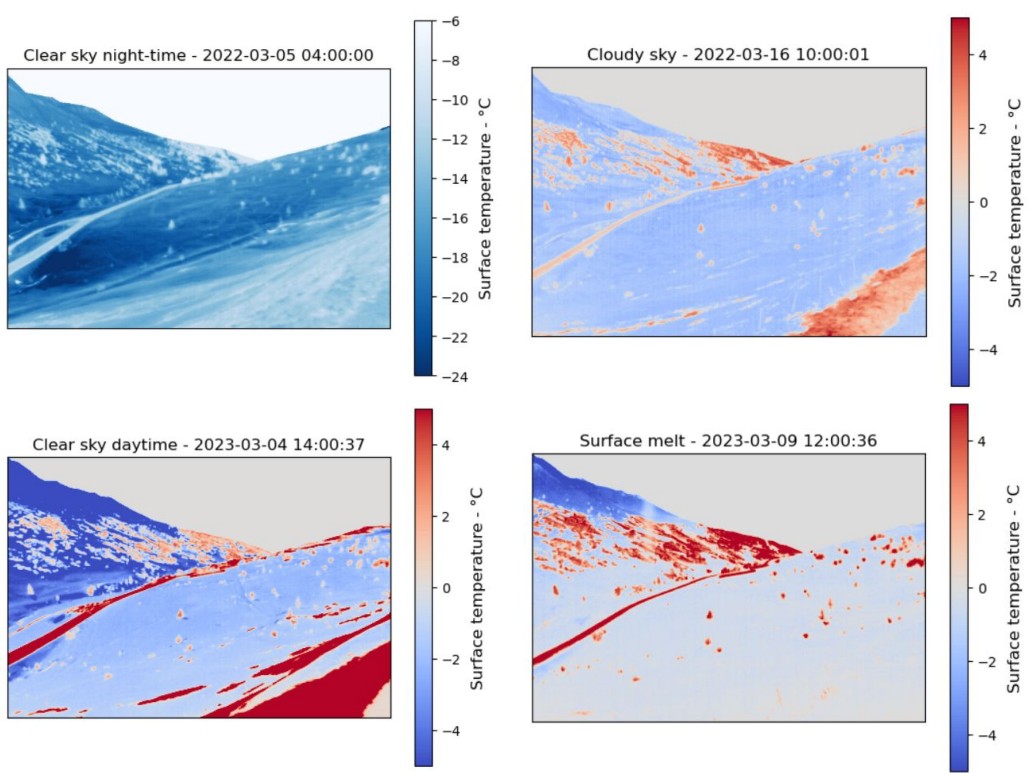

**Figure 14.** Four measurements of the snow surface temperature acquired using a TIR camera under different atmospheric conditions.

## 10 Data availability

Snow surface radiative temperature maps and ancillary data can be found at https://doi.org/10.57932/8ed8f0b2-e6ae-4d64-97e5-1ae23e8b9

(Arioli et al., 2024a) and https://doi.org/10.57932/1e9ff61f-1f06-48ae-92d9-6e1f7df8ad8c (Arioli et al., 2024b). The data is
partitioned in two datasets, one for winter 2021-2022, the other for spring 2023. Both contains maps of the surface temperature
of snow acquired at the Col du Lautaret every 2 minutes, RGB images of the scene acquired every 10 minutes during operation
of the TIR camera, timeseries of the surface temperature acquired every 30 seconds at AWS-North and AWS-South, the TIR
camera's internal temperature acquired every 2 seconds. The dataset from spring 2023 also includes timeseries acquired every

30 seconds at AWS-North and AWS-South of the incoming longwave radiation, air temperature, air humidity and wind speed
(AWS-South only). Each dataset contains zip files corresponding to the week of the year of acquisition (starting each Mon-
day). For example 2023_w06.zip refers to week 6 of the year 2023. Every .zip file contains the folders with the data separated
by day. Within each day folder, the *maps_Ts* folder contains the surface temperature maps single-band in geotiff format, in
the EPSG:32632 projection and LWZ compression. They are 464x506 pixels wide at 2 m resolution, with float32 data. The

*RGB* folder contains the visible images of the scene. The file *Camera_Tint.csv* contains the timeseries of the internal temper-

ature of the TIR camera. The files *AWS_north.csv* and *AWS_south.csv* contain timeseries of data measured at AWS-North and AWS-South respectively. More details are in the *README.md* file in each dataset.

**Appendix A**

Table A1 and Table A2 show the positions of the GCPs used for the orthorectification of the TIR imagery. They are 21 for winter 2021-2022 and 25 for spring 2023. Their positions are supplied both in world and camera coordinates, and the error projection for each GCP is given. Fig. A1 and Fig. A2 show the spatial distribution of the GCPs for winter 2021-2022 and for spring 2023 respectively. The distribution is shown both in camera coordinates and longitude-latitude (EPSG:4326).

| GCP | X | Y | Longitude | Latitude | Projection error (m) |
|---|---|---|---|---|---|
| 1 | 257.4 | 327.4 | 6.3979958 | 45.0339904 | 6.82 |
| 2 | 223.5 | 269.4 | 6.3965944 | 45.0331135 | 2.92 |
| 3 | 301.6 | 264.7 | 6.3965362 | 45.0336515 | 2.52 |
| 4 | 402.1 | 293.9 | 6.3972586 | 45.0344586 | 0.37 |
| 5 | 463.7 | 246.5 | 6.3960861 | 45.0345687 | 10.23 |
| 6 | 164.8 | 345.4 | 6.3983713 | 45.0337279 | 4.73 |
| 7 | 294.2 | 245.5 | 6.3953553 | 45.0331503 | 0.12 |
| 8 | 345.7 | 235.7 | 6.3949846 | 45.0334695 | 2.83 |
| 9 | 365.3 | 226.3 | 6.3945059 | 45.0335157 | 7.31 |
| 10 | 477.6 | 204.4 | 6.3939576 | 45.0343643 | 5.58 |
| 11 | 32.0 | 277.2 | 6.3970523 | 45.0316694 | 4.57 |
| 12 | 13.2 | 416.8 | 6.3992936 | 45.0337638 | 3.08 |
| 13 | 510.3 | 214.7 | 6.3948275 | 45.0347281 | 4.24 |
| 14 | 128.3 | 323.2 | 6.3982064 | 45.0334006 | 5.92 |
| 15 | 171.3 | 284.7 | 6.3972699 | 45.0330215 | 3.91 |
| 16 | 138.2 | 302.2 | 6.3976477 | 45.0330839 | 13.72 |
| 17 | 284.2 | 307.4 | 6.3976468 | 45.0339967 | 0.19 |
| 18 | 578.5 | 179.0 | 6.3933326 | 45.0351565 | 2.90 |
| 19 | 81.4 | 287.5 | 6.3967788 | 45.0319156 | 0.01 |
| 20 | 278.4 | 347.8 | 6.3982973 | 45.0342323 | 1.70 |
| 21 | 373.3 | 239.3 | 6.3955060 | 45.0338575 | 5.92 |

**Table A1.** List of the ground control points (GCPs) used for the orthorectification of the thermal infrared images acquired during winter 2021-2022. Both the position of the GCPs in the camera coordinates (X, Y) and world coordinates (longitude and latitude, EPSG:4326) are supplied. The last column indicates the error projection computed by the Ames Stereo Pipeline during the triangulation of the GCPs.

| GCP | X | Y | Longitude | Latitude | Projection error (m) |
|---|---|---|---|---|---|
| 1 | 83.9 | 260.7 | 6.3964142 | 45.0314624 | 0.96 |
| 2 | 116.8 | 242.8 | 6.3952909 | 45.0309603 | 5.00 |
| 3 | 141.3 | 233.8 | 6.3945973 | 45.0307771 | 2.53 |
| 4 | 187.5 | 237.8 | 6.3939559 | 45.0310169 | 1.27 |
| 5 | 204.5 | 251.3 | 6.3946054 | 45.0316481 | 0.97 |
| 6 | 187.7 | 256.2 | 6.3953463 | 45.0317810 | 0.88 |
| 7 | 142.2 | 288.7 | 6.3965102 | 45.0321692 | 2.41 |
| 8 | 131.4 | 291.7 | 6.3970462 | 45.0324629 | 3.45 |
| 9 | 152.1 | 290.5 | 6.3974828 | 45.0329644 | 0.45 |
| 10 | 184.3 | 290.8 | 6.3974261 | 45.0331668 | 4.85 |
| 11 | 151.4 | 317.1 | 6.3981059 | 45.0333973 | 1.07 |
| 12 | 117.6 | 369.1 | 6.3988032 | 45.0337287 | 1.98 |
| 13 | 224.0 | 373.4 | 6.3986794 | 45.0341295 | 6.09 |
| 14 | 274.1 | 324.3 | 6.3977354 | 45.0339391 | 6.90 |
| 15 | 241.3 | 332.4 | 6.3981241 | 45.0339068 | 1.00 |
| 16 | 88.0 | 200.1 | 6.3946323 | 45.0300613 | 10.13 |
| 17 | 47.3 | 270.4 | 6.3970523 | 45.0316694 | 0.51 |
| 18 | 96.5 | 284.5 | 6.3967788 | 45.0319156 | 3.43 |
| 19 | 27.1 | 406.2 | 6.3992936 | 45.0337638 | 3.31 |
| 20 | 307.5 | 245.5 | 6.3953553 | 45.0331503 | 5.49 |
| 21 | 358.2 | 237.8 | 6.3949846 | 45.0334695 | 5.83 |
| 22 | 297.9 | 307.9 | 6.3976468 | 45.0339967 | 1.76 |
| 23 | 289.0 | 346.5 | 6.3982973 | 45.0342323 | 3.21 |
| 24 | 386.7 | 243.3 | 6.3955060 | 45.0338575 | 5.83 |
| 25 | 471.1 | 292.7 | 6.3971422 | 45.0347195 | 1.86 |

**Table A2.** List of the ground control points (GCPs) used for the orthorectification of the thermal infrared images acquired during spring. Both the position of the GCPs in the camera coordinates (X, Y) and world coordinates (longitude and latitude, EPSG:4326) are supplied. The last column indicates the error projection computed by the Ames Stereo Pipeline during the triangulation of the GCPs.

*Author contributions.* SA, GP, SG, LA and MI designed the study. LA, SA and GP performed the installation and maintenance of instrumentation on the field. SA, LA and EAG performed the fieldwork. SA characterized the instrument, implemented the processing of the

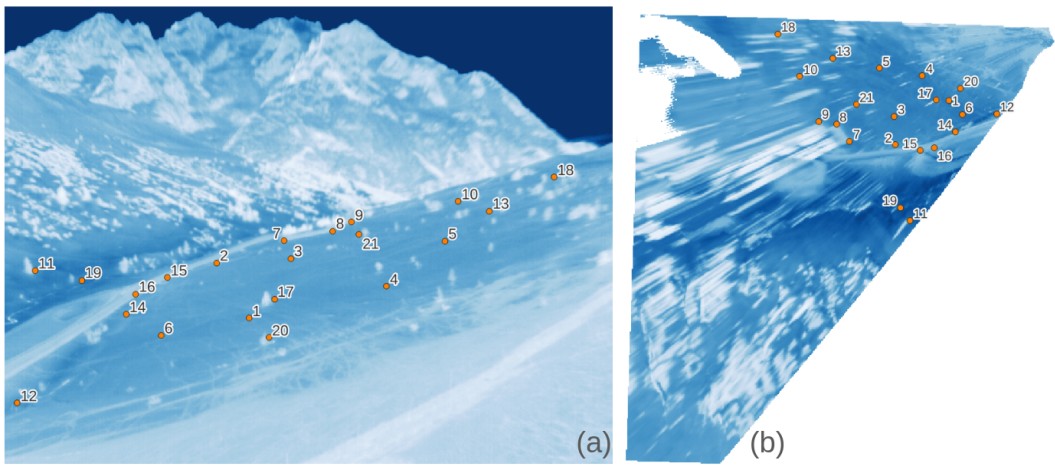

**Figure A1.** Spatial distribution of the ground control points (GCPs) listed in Table A1 in camera coordinates (a) and world coordinates (longitude and latitude, b).

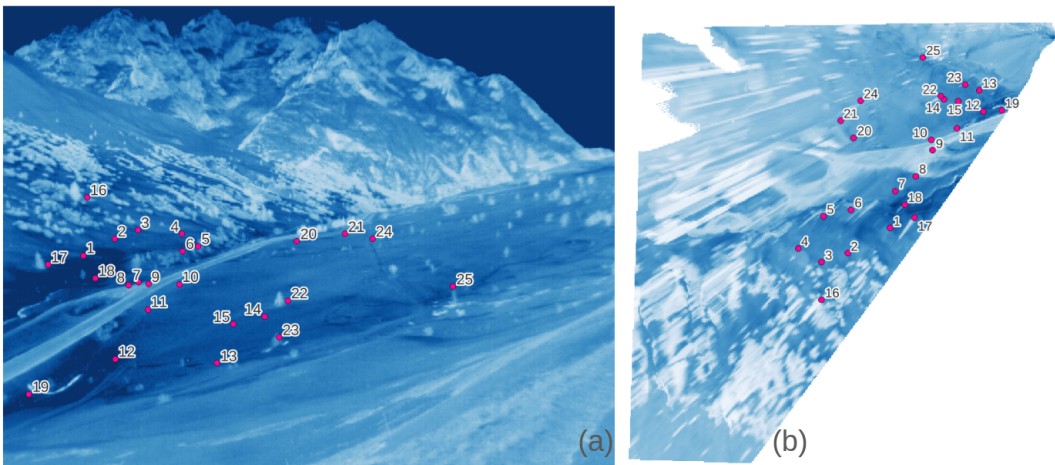

**Figure A2.** Spatial distribution of the ground control points (GCPs) listed in Table A2 in camera coordinates (a) and world coordinates (longitude and latitude, b).

observations, performed the analysis and wrote the manuscript. EAG and MP contributed to the geometric processing of the images. All authors discussed and revised the manuscript.

*Competing interests.* All authors declare no competing interests.

*Acknowledgements.* This study has been supported by the Centre National d'Etudes Spatiales (TRISHNA), the European Space Agency (4D Antarctica) and the Région Auvergne-Rhône-Alpes (SENSASS). It has benefited from the use of the infrastructures of the Lautaret Garden
– UAR 3370 (Univ. Grenoble Alpes, CNRS, 38000 Grenoble, France), a member of AnaEE France (ANR-11-INBS- 0001 AnaEE-Services, Investissements d'Avenir frame) and of the eLTER European network (Univ. Grenoble Alpes, CNRS, LTSER Zone Atelier Alpes, 38000 Grenoble, France). The authors would like to thank the members of the UMR 3370 - Jardin du Lautaret - UGA/CNRS for their indispensable logistic help and involvement during field campaigns.

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
