# Peer review of "Time series of alpine snow surface radiative temperature maps from high precision thermal infrared imaging"

_Earth System Science Data, 2024_

## Referee Comment (RC1)

**Manuscript Review:**

**Time series of alpine snow surface radiative temperature maps from high precision thermal infrared imaging.**

**Arioli et al., 2024**

The manuscript presents a dataset comprised of a series of thermal maps/images and RGB images collected from a single observation station looking across a snow-covered mountain area in the French Alps. The paper presents perhaps the most rigorous and best example of calibration routines for thermal imaging over snow surfaces (and perhaps for any land surface thermal imaging) in the current literature. The authors address most of the major sensor limitations of microbolometer thermal sensors; including sensor drift, internal temperature, bias correction, emissivity, etc. This is a technically very challenging task and there are many valuable insights in the workflow presented that will be of great use to the research community. The datasets themselves are likely to provide a useful resource for snow studies and calibration of future satellite missions (if the dataset continues to be collected contemporaneously with these), alongside being a useful dataset in their own right for studies of snow surface energy balance, heterogeneity, etc. Furthermore, the manuscript is well written, and easy to follow.

However, I do have one major comment on the paper. While the authors account for most of the error sources of microbolometer sensors they do not address the potential impact of variable viewing angles and variable distance to target on their measurements/dataset. Studies have shown these can both potentially significantly impact thermal camera measurements. The impact of these on the datasets is likely not apparent in the error assessment against the more accurate IR sensors on the AWS because these are all located at roughly similar distances from the camera and have similar viewing angles. As a result, I believe the reported absolute error values of 0.67K (spring 2023) is probably only representative of areas with similar distance and viewing angle to the AWS stations and is likely worse in other areas of the study site, these limitations should ideally be investigated to confirm or negate their presence. If present the datasets could be improved by addressing these issues, however this is likely to be fairly labour intensive and the dataset and manuscript in their current form are still at the forefront of research in this field. To produce the most accurate and thus useful dataset I suggest the authors identify and address these error sources, however given the high quality of the paper and datasets already I would be satisfied if their potential impacts and consequent limitations were discussed within the paper, at a minimum.

**Major comments:**

*Viewing Angle:*

Other studies have shown that microbolometer sensors suffer a significant reduction in measured temperatures when the viewing angle is greater than ~40° (Cheng et al., 2012; Jiao et al., 2016; Litwa, 2010). This is potentially what is visible in Figure 5 of the manuscript at the periphery of the black body where viewing angle increases, though this could also be temperature bleeding across the boundary of the calibration surface and the background. When shooting thermal imagery from a nadir position (e.g. from a drone/plane) the viewing angle is generally less than 40° (unless flying over very complex topography) and this effect is minimised. However, when shooting an oblique as is the case in this manuscript much of the image is taken from angles exceeding 40°, when the land is

sloping away (blue ellipses) from the camera this effect is enhanced, while when the land faces towards (red ellipses) the camera the effect is minimised. These errors are likely not picked up in the bias correction to the AWS stations because they all appear to have a very similar viewing angle from the camera. I would suggest first testing the sensor to see if, and to what extent it suffers from viewing angle errors (which is likely). If this is confirmed then these errors could be corrected by calculating the impact of viewing angle on their sensor in a lab using a black body of known temperature (as shown in Figure 5 of the manuscript) e.g. following Alexander & Lunderman, (2021) and then applying this correction to the image temperature measurements after calculating each pixels view angle from the camera.

**Viewing Distance:**

The secondary issue is distance to sensor (Minkina & Klecha, 2016). Microbolometers are impacted by distance to the target for a number of reasons, and this also varies somewhat depending on atmospheric conditions at the time of imaging. When images are collected from a nadir position (e.g. drone/plane) this variability is minimal as height above ground can be kept relatively similar/constant (except with very complex topography). With oblique imagery this is not the case unless the target of interest is confined to a single location/distance to sensor. The manuscript presents data that is spread horizontally across ~1.2km of ground from the sensor/camera to the area beyond the AWS (but excluding the very distant mountains). There is likely a significant trend in the dataset because of this. The more accurate sensors on the AWS are all located at a roughly similar distance from the camera (~700m) and therefore are unlikely to pick up this effect/trend. To coarsely remedy this the authors could work out their sensors distance to target sensitivity in a lab by collecting images of a stable black body calibration plate from known distances (though extrapolating measurements from a few metres in the lab to many km in the field may be dubious and). Some cameras also provide this information within their software as a distance to target correction factor. The correction factor could then be applied to all the pixels dependent on their distance from the camera. This approach would ignore the impact of variations in atmospheric conditions between scenes and across the ~1.2km distance, however this is likely much too hard to correct for.

[Figure]

[Figure]

Blue areas slope steeply away from sensor, view angle likely >60. Approx. 0-200m away from sensor.

Yellow areas slope slightly towards sensor, view angle potentially >40°. Approx. 300-800m away from sensor.

Red areas slope towards sensor, view angle potentially <40°. Approx. 600-1200m away from sensor.

**Minor Comments:**

AWS coordinates are provided in the readme of the datafile in decimal degrees to 4 decimal places. This means >10-13m precision (latitude dependent). Given that the TIR map product is at 2m spatial resolution, it would be better to have the AWS coordinates to 5 or 6 decimal places (~1m to 0.1m precision) so that the field of view of the AWS IR sensors can be matched to the correct pixel within the scene. Ideally the actual centre point of what the IR sensor is looking at should also be provided as a coordinate in the readme file, not just the AWS/tower coordinates which I think are what is currently provided.

It would be useful for users to also have a georectified version of the RGB imagery made available so that users can directly link thermal maps to ground surface conditions at the pixel scale, but this is not 100% necessary and is likely to require more effort to produce.

Line 1 temperature of 'the' snow cover – delete 'the'

Line 26 improve – change to improving?

Line 56  cal/val change to calibration-validation as used elsewhere in the manuscript.

Line 111 'in a small volume'. Here and in later sentences I think the word volume is referring to the secondary external case that the camera was placed inside (and then temperature regulated). Change volume to case/box/enclosure or something to that effect for clarity.

Line 120/121 does this refer to the temperature setting of the volume/enclosure itself? How does raising the internal temperature by 7°C (15 raised to 22C) reduce overheating? Surely that would increase overheating?  Check and revise if required.

141 A map showing the GCP locations could be useful, to assess the distribution and likely accuracy of this – some details on the accuracy of the georectification would also be beneficial here. I presume the GCPs were located in areas that were not snow covered during the winter and are constant and visible in every image? Also providing these GCP coordinates so that others can orthorectify the RGB imagery if it is not to be provided in a map format would be useful. I'm also wondering how well the 100x70cm GCPs were visible in the TIR imagery which is provided at a 2m spatial resolution. If the GCPs are well distributed across the survey area and can be assumed to accurately reflect sky temperature only (i.e. constant and stable for a given image) then perhaps they could be useful for identifying potential errors in Ts resulting from view angle and distance to target.

Table 2: lat long are presented to 6 sig fig precision (5dp longitude, 4dp latitude). These should be at least 5dp precision (ideally 6dp) in all cases so we are at ~metre scale accuracy at minimum.

Figure 11: a density scatterplot might better show the distribution of this many points. I think the black line is the 0/0 line and not the $R^2$ linear fit? Please add linear fit along with $R^2$ and RMSE. From the 2023 figure it looks like there is a warm bias in the camera measurement that increases with surface temperature (excluding the section circled in red), it might be worth discussing this too.

Line 346 – I have also noticed this warm temp in my own work of TIR imaging of melting snow/ice where I would expect 0°C. I wonder if it could be due to the camera actually measuring the temperature of a thin film of melt water sitting atop the snow/ice which could potentially exceed the 0°C of the snow/ice surface proper?

Alexander, Q. G., & Lunderman, C. V. (2021). *ERDC/ITL TN-21-1 "Thermal camera reliability study : FLIR One Pro."* Retrieved April 16, 2024, from https://erdc-library.erdc.dren.mil/jspui/handle/11681/42180

Cheng, T. Y., Deng, D., & Herman, C. (2012). CURVATURE EFFECT QUANTIFICATION FOR IN-VIVO IR THERMOGRAPHY. *International Mechanical Engineering Congress and Exposition : [Proceedings]. International Mechanical Engineering Congress and Exposition*, *2*, 127–133. https://doi.org/10.1115/IMECE2012-88105

Jiao, L., Dong, D., Zhao, X., & Han, P. (2016). Compensation method for the influence of angle of view on animal temperature measurement using thermal imaging camera combined with depth image. *Journal of Thermal Biology*, *62*, 15–19. https://doi.org/10.1016/J.JTHERBIO.2016.07.021

Litwa, M. (2010). Influence of angle of view on temperature measurements using thermovision camera. *IEEE Sensors Journal*, *10*(10), 1552–1554. https://doi.org/10.1109/JSEN.2010.2045651

Minkina, W., & Klecha, D. (2016). Atmospheric transmission coefficient modelling in the infrared for thermovision measurements. *J. Sens. Sens. Syst*, *5*, 17–23. https://doi.org/10.5194/jsss-5-17-2016

---

## Referee Comment (RC2)

[Figure]

**Figure 1**. RGB image on left showing the location of this stand of trees. TIR image on right showing the result of projecting the image onto a bare-ground DEM where the trees are then draped over the terrain behind them, causing a large area to appear the same temperature as the sides of these trees.

[Figure]

**Figure 2**. At the same location as the stand of trees in Figure 1, when a car is captured in an image passing by this spot its warm signature is also draped across the terrain behind the car. This causes narrow warm streaks in some images. (Cars are visible elsewhere on the road in other images where this effect is minimal due to a different view angle)

---

## Author Comment (AC1)

The authors would like to thank the reviewers for their comments and feedback. Our answers are presented below in blue. For the minor comments, the check mark indicates that the suggested modification was implemented.

**Major comments**

**Viewing Angle**: Other studies have shown that microbolometer sensors suffer a significant reduction in measured temperatures when the viewing angle is greater than 40° (Cheng et al., 2012; Jiao et al., 2016; Litwa, 2010). This is potentially what is visible in Figure 5 of the manuscript at the periphery of the black body where viewing angle increases, though this could also be temperature bleeding across the boundary of the calibration surface and the background. When shooting thermal imagery from a nadir position (e.g. from a drone/plane) the viewing angle is generally less than 40° (unless flying over very complex topography) and this effect is minimised. However, when shooting an oblique as is the case in this manuscript much of the image is taken from angles exceeding 40°, when the land is sloping away (blue ellipses) from the camera this effect is enhanced, while when the land faces towards (red ellipses) the camera the effect is minimised. These errors are likely not picked up in the bias correction to the AWS stations because they all appear to have a very similar viewing angle from the camera. I would suggest first testing the sensor to see if, and to what extent it suffers from viewing angle errors (which is likely). If this is confirmed then these errors could be corrected by calculating the impact of viewing angle on their sensor in a lab using a black body of known temperature (as shown in Figure 5 of the manuscript) e.g. following Alexander & Lunderman, (2021) and then applying this correction to the image temperature measurements after calculating each pixels view angle from the camera.

According to our literature review, the variations of measured IR temperature due to the viewing angle depend on three possible elements:

1. the angular response of the sensors, the microbolometers. In this study the variation of the angular response is taken into account by the manufacturer's calibration of the instrument, and if this were not sufficient, the Flat Field Correction (FFC) would correct any potential residual angular response of the microbolometers.

2. vignetting, i. e. reduced luminosity in the periphery of the image due to the lens properties. This problem is also corrected by the FFC.

3. the directional emissivity of the target, as e.g. discussed in [Litwa(2010)]. As this issue was raised by both reviewers, the answer in the following paragraph is common to the two reviews.

As pointed out, the emissivity of snow is likely decreasing with the angle of observation. However, only few measurements of the directional snow emissivity are available in literature. These observations suggest that snow cover emissivity is mostly isotropic in contrast with most natural surfaces. However, angular variations were observed depending on several factors including density, age and grain size and there is no consensus on how to model them. To avoid an inappropriate correction, our approach is to consider that this effect contributes to the global error, that is assessed by the comparison with the ground truth measurements (that are not affected by the grazing incidence of the camera).

Nonetheless, to facilitate further investigation, as suggested by the reviewer, we add a map of the observation angles of the snow surface temperature in the published dataset.

To clarify this, the paragraph at L350 in the Discussion was modified as follows: "Another limitation of the current dataset concerns the emissivity. We assume snow is a perfect black-body and do not attempt to apply an emissivity model due to the lack of consensus on the topic. However, snow emissivity is known to be slightly lower than 1 (Hori et al. 2014) and to decrease with the observation angle, especially for angles over 45° in the 12-14 $\mu$m TIR window (Dozier and Warren, 1982, Hori et al., 2006). We assume this error is included in the global error, assessed by the comparison with the ground truth. To facilitate further analysis by the data users, a map of the observation angles of the area, computed as in Castel et al. (2001), is included in the dataset. Moreover, downwelling longwave measurements performed at the Col du Lautaret site are supplied (FluxAlp measurements for winter 2021-2022, AWS-North and AWS-South measurements for spring 2023). With these measurements, the surface radiative temperatures can be adjusted to accommodate emissivity values < 1".

We would also like to specify that the cold aura visible in Fig. 5 of the study is not caused by any of the three effects described above. Indeed, the black body used for the calibration - Landcal P80P has a diameter 50 mm and is located at the end of a 50 mm wide, 160 mm long cavity. It was therefore not possible to approach the TIR camera close enough to the black body source in order for it to cover the whole field of view. We therefore estimate that the gradually changing temperature at the periphery of Fig. 5 represents the cavity temperature, gradually varying towards the ambient temperature to which the casing of the black body source is exposed. In order to make this clearer in the manuscript we suggest the following addition at L189: "The Landcal P80P black-body source has a diameter 50 mm and is located at the end of a 50 mm wide and 160 mm long cavity. It was therefore not possible to approach the TIR camera close enough to the black body source to match the field of view and the area of homogeneous temperature. The calibration assessment was thus carried out using the central 240x210 pixel rectangle shown in Fig. 5." The caption of Fig. 5 was also changed accordingly.

**Viewing Distance**: The secondary issue is distance to sensor (Minkina & Klecha, 2016). Microbolometers are impacted by distance to the target for a number of reasons, and this also varies somewhat depending on atmospheric conditions at the time of imaging. When images are collected from a nadir position (e.g. drone/plane) this variability is minimal as height above ground can be kept relatively similar/constant (except with very complex topography). With oblique imagery this is not the case unless the target of interest is confined to a single location/distance to sensor. The manuscript presents data that is spread horizontally across 1.2km of ground from the sensor/camera to the area beyond the AWS (but excluding the very distant mountains). There is likely a significant trend in the dataset because of this. The more accurate sensors on the AWS are all located at a roughly similar distance from the camera ( 700m) and therefore are unlikely to pick up this effect/trend. To coarsely remedy this the authors could work out their sensors distance to target sensitivity in a lab by collecting images of a stable black body calibration plate from known distances (though extrapolating measurements from a few metres in the lab to many km in the field may be dubious and). Some cameras also provide this information within their software as a distance to target correction factor. The correction factor could then be applied to all the pixels dependent on their distance from the camera. This approach would ignore the impact of variations in atmospheric conditions between scenes and across the 1.2km distance, however this is likely much too hard to correct for.

As pointed out by [Minkina and Klecha(2016)], the measured temperature is influenced by the radiative contribution of the air between the sensor and the target. However, we did not account for this effect and consider it part of the measurement error when comparing with the ground truth at the two AWS that are much less affected by this issue. This assumption is based on the fact that the camera spectral range (8-14 µm) corresponds to the infrared atmospheric window where there is little absorption by atmospheric gases. We assess this assumption using a case during the melt season where we expect that the snow surface temperature should be homogeneous at 0°C across the landscape. In Fig. 1 we plot the surface temperature retrieved from the camera dataset along a transect (on the left) against the surface to the camera distance (on the right). We find that there is no trend (within our target uncertainty of ± 0.67K) which corroborates our assumption that the atmosphere opacity is negligible in this case ($T_{air}$=4.6°C, $RH$=57% at AWS-South). However, this assumption may not hold at times of higher air absolute humidity. Therefore, we agree with the reviewer that it is important to inform potential users about the potentially increasing error with the viewing distance due to atmospheric effects. For this, we add a distance-to-camera map to the dataset, that can be used with temperature and humidity measurements from the AWS to assess and potentially correct for the atmospheric contribution. Moreover, we add the following sentences to the manuscript:

1. In the Processing section, L216: "Our method does not include corrections for non-instrumental effects such as emissivity and atmospheric contributions. We therefore consider these effects to be included in our error estimations".

2. In the Discussion, L355: "Similarly, the measured temperature depends on the radiative contribution of the air between the sensor and the target, that was not corrected and is therefore considered part of the error measured in the comparison with the ground truth at the two AWS. Still, as this error increases with the viewing distance, a distance-to-camera map is included in the dataset, and can be used with temperature and humidity measurements from the AWS to correct for the atmospheric contribution".

[Figure]

Figure 1: Snow-covered transect (on the left, yellow line) on the surface temperature map acquired on the 17-04-2024 at 14:01:01, and the measured temperature as a function of distance from the TIR camera (on the right).

**Minor Comments**

- AWS coordinates are provided in the readme of the datafile in decimal degrees to 4 decimal places. This means 10-13m precision (latitude dependent). Given that the TIR map product is at 2m spatial resolution, it would be better to have the AWS coordinates to 5 or 6 decimal places (1 m to 0.1 m precision) so that the field of view of the AWS IR sensors can be matched to the correct pixel within the scene. Ideally the actual centre point of what the IR sensor is looking at should also be provided as a coordinate in the readme file, not just the AWS/tower coordinates which I think are what is currently provided. ✓

- It would be useful for users to also have a georectified version of the RGB imagery made available so that users can directly link thermal maps to ground surface conditions at the pixel scale, but this is not 100% necessary and is likely to require more effort to produce.
  We agree that an orthorectified version of the RGB images could enhance the comprehension of the TIR imagery. However, the camera is still installed on the field, which made it impossible to test it for distortion parameters. As a result, our attempts to orthorectify the RGB images have not been successful.

- Line 1: temperature of 'the' snow cover – delete 'the' ✓

- Line 26: improve – change to improving? ✓

- Line 56: cal/val change to calibration-validation as used elsewhere in the manuscript. ✓

- Line 111 'in a small volume'. Here and in later sentences I think the word volume is referring to the secondary external case that the camera was placed inside (and then temperature regulated). Change volume to case/box/enclosure or something to that effect for clarity. ✓

- Line 120/121: does this refer to the temperature setting of the volume/enclosure itself? How does raising the internal temperature by 7°C (15 raised to 22C) reduce overheating? Surely that would increase overheating? Check and revise if required.
  Line 120/121 do refer to the temperature setting of the enclosure including the TIR camera. Indeed, the TEM cools off the small enclosure within the camera's casing by warming the back of the casing, where heat is expelled towards the outside through radiators installed in contact with its conductive, metallic walls. However, heat evacuation in this setup is limited and, if the external temperature is significantly higher than the TEM setting temperature, heat evacuation from the rear of the casing might become impossible. If this happens, the TEM would cause further overheating while trying to cool down the camera. Reducing the difference between the internal and external temperature - by raising the TEM setting temperature from 15°C to 22°C - reduces overheating episodes because it reduces the required heat evacuation. To make

this clearer in the manuscript, we modify the paragraph at L120/121 as follows: "The internal temperature of the camera was initially set to 15°C. However, this temperature proved to be too low with respect to the limited heat evacuation in the rear of the casing of this setup, causing significant overheating when the external temperature approaches 15°C. It was then raised to 22°C on the 2023-02-17 after several episodes of overheating occurred during the first week of measurements. By raising the TEM setting temperature from 15°C to 22°C, the required heat evacuation was reduced and no overheating episodes happened during the rest of the season".

- Line 141: A map showing the GCP locations could be useful, to assess the distribution and likely accuracy of this – some details on the accuracy of the georectification would also be beneficial here. I presume the GCPs were located in areas that were not snow covered during the winter and are constant and visible in every image? Also providing these GCP coordinates so that others can orthorectify the RGB imagery if it is not to be provided in a map format would be useful. I'm also wondering how well the 100x70cm GCPs were visible in the TIR imagery which is provided at a 2m spatial resolution. If the GCPs are well distributed across the survey area and can be assumed to accurately reflect sky temperature only (i.e. constant and stable for a given image) then perhaps they could be useful for identifying potential errors in Ts resulting from view angle and distance to target.

  A table containing the list of GCPs and the projection error computed by the Ames Stereo Pipeline software for the years 2022 and 2023 was added to the manuscript in the Appendix A. The following sentences are added to the manuscript:

  - L145: "Their positions both in world and camera coordinates are listed in Table A1 and Table A2 and shown in Fig. A1 and Fig. A2 in Appendix A."
  - L300: "The projection error for each GCPs is listed in Table A1 and Table A2 in Appendix A."

  The GCPs' positions were indeed acquired during the winter season for both years, with snow covering the ground. To compensate for the presence of the snow cover, 0.75 m were added to the DEM used for the orthoprojection. According to the distance from the TIR camera, the aluminum plate used to extract the GCPs position is identified as a 1 to 2 pixel cold spot appearing at the time of acquisition. However, because of its small size, the aluminum plate is not completely resolved in the image and we cannot use its temperature as a measurement for the sky temperature.

- Table 2: lat long are presented to 6 sig fig precision (5dp longitude, 4dp latitude). These should be at least 5dp precision (ideally 6dp) in all cases so we are at metre scale accuracy at minimum. ✓

- Figure 11: a density scatterplot might better show the distribution of this many points. I think the black line is the 0/0 line and not the R2 linear fit? Please add linear fit along with R2 and RMSE. From the 2023 figure it looks like there is a warm bias in the camera measurement that increases with surface temperature (excluding the section circled in red), it might be worth discussing this too.

  As suggested, Figure 11 was replaced by two density scatterplots. The linear regressions were computed and the slope, intercept, RMSE and $R^2$ of each are listed. The caption is modified as: "Density distributions of the snow surface temperatures measured at AWS-South by the camera as a function of those measured by the radiometer during winter 2021-2022 (a), and spring 2023 (b). The plot shows the high measurements of the surface temperature measured during spring 2023 over melting snow (red circle). Also shown are the identity line (dotted gray) and the linear regression of the distributions (orange)." Moreover, the following paragraph is added to the end of the Validation section: "Also illustrated in Fig. 11 are the linear regressions of the two distributions (orange lines), the computed slopes, intercepts, fit $R^2$ and RMSE (respectively $m$, $b$, $R^2$ and $\sigma_y$). Both distributions confidently fit to the regression, as shown by the elevated $R^2$, 0.96 for winter 2021-2022, 0.97 for spring 2023. The first has an intercept $b$=-1.81°C with the error decreasing with the temperature as the slope $m$=0.91. The second, on the other hand, has an intercept $b$=+0.52°C and the bias is fairly constant as $m$=1.02."

- Line 346 – I have also noticed this warm temp in my own work of TIR imaging of melting snow/ice where I would expect 0°C. I wonder if it could be due to the camera actually measuring the temperature of a thin film of melt water sitting atop the snow/ice which could potentially exceed the 0°C of the snow/ice surface proper?
  We have evaluated this hypothesis. Assuming a thermal conductivity of water $k$=0.6 and a water film thickness of 0.5 mm, 1200 $Wm^{-2}$ of absorbed energy would be needed to sustain a 1 K difference between the top layer and the bottom snow layer at 0°C, which is not plausible. In order to clarify this aspect, we modify L346 as follows: "We tested the possibility that the higher error for melting snow might be due to a thin layer of warmer water above the snow grains. However, assuming a thermal conductivity of water $k$=0.6 and a water film thickness of 0.5 mm, 1200 $Wm^{-2}$ of absorbed energy would be needed to sustain a 1 K difference between the top layer and the bottom snow layer at 0°C, which is not plausible. Therefore, the physical cause of this error is unidentified and remains to investigate."

Finally, although not requested by the reviewers, two figures were added to the manuscript to highlight the utility of the dataset. The first is a full timeseries of the snow surface temperature measured at AWS-South from between December 2022 and May 2023, compared with the TIR radiometer. The second illustrates four examples of spatial patterns of the snow surface temperature under different atmospheric conditions: a night with clear sky, a day with clear sky, a cloudy day and a melting day. To implement these figures, L376 was modified as follows: "The full timeseries are shown in Fig. 13. Finally, the images in Fig. 14, acquired under different atmospheric conditions, show the variability of the patterns of the surface temperature of snow caught by the TIR camera. Beside these examples, the dataset provides numerous other situations that deserve further exploration."

**References**

[Litwa(2010)] Litwa, M.: Influence of angle of view on temperature measurements using thermovision camera, IEEE Sensors Journal, 10, 1552–1554, https://doi.org/10.1109/JSEN.2010.2045651, 2010.

[Minkina and Klecha(2016)] Minkina, W. and Klecha, D.: Atmospheric transmission coefficient modelling in the infrared for thermovision measurements, Journal of Sensors and Sensor Systems, 5, 17–23, https://doi.org/10.5194/jsss-5-17-2016, 2016.

---

## Author Comment (AC2)

The authors would like to thank the reviewers for their comments and feedback. Our answers are presented below in blue.

**Major comments**

**Emissivity**: The authors note the high, near-blackbody, emissivity of snow surfaces with some variation due to grain size, but neglect to address the angular variability of snow emissivity (see Dozier & Warren 1982, which the authors did cite). The difference between observed temperature using a blackbody assumption and the true surface temperature of snow is likely much larger than the errors from other sources that the authors did investigate (e.g. window transmissivity). High spatial resolution DEMs of the study site were used in the projection of the imagery, therefore I suggest that the authors use these same DEMs to estimate snow emissivities given the varying view angles across the study site. Alternatively, an additional map could be provided with the dataset to show the variation in view angle across the study area so that a user of the data can apply their own angular emissivity corrections across the image.

As this comment was raised by both reviewers, the following answer is common to the two reviews. As pointed out, the emissivity of snow is likely decreasing with the angle of observation. However, only few measurements of the directional snow emissivity are available in literature. These observations suggest that snow cover emissivity is mostly isotropic in contrast with most natural surfaces. However, angular variations were observed depending on several factors including density, age and grain size and there is no consensus on how to model them. To avoid an inappropriate correction, our approach is to consider that this effect contributes to the global error, that is assessed by the comparison with the ground truth measurements (that are not affected by the grazing incidence of the camera).

Nonetheless, to facilitate further investigation, as suggested by the reviewer, we add a map of the observation angles of the snow surface temperature in the published dataset.

To clarify this, the paragraph at L350 in the Discussion was modified as follows: "Another limitation of the current dataset concerns the emissivity. We assume snow is a perfect black-body and do not attempt to apply an emissivity model due to the lack of consensus on the topic. However, snow emissivity is known to be slightly lower than 1 (Hori et al. 2014) and to decrease with the observation angle, especially for angles over 45° in the 12-14 $\mu$m TIR window (Dozier and Warren, 1982, Hori et al., 2006). We assume this error is included in the global error, assessed by the comparison with the ground truth. To facilitate further analysis by the data users, a map of the observation angles of the area, computed as in Castel et al. (2001), is included in the dataset. Moreover, downwelling longwave measurements performed at the Col du Lautaret site are supplied (FluxAlp measurements for winter 2021-2022, AWS-North and AWS-South measurements for spring 2023). With these measurements, the surface radiative temperatures can be adjusted to accommodate emissivity values < 1".

**Image projection**: Inspecting the TIR imagery, I noticed an area in which there were significant parallax effects due to the images of trees being projected onto a large area of adjacent terrain (see attached Figure 1). It appears that this line of trees (behind a snow fence from the camera's point of view) occupies a slight rise in terrain before the terrain dips into a ravine or stream valley below. When the TIR images are projected using a bare-ground DEM, the sides of these trees are therefore draped across a large swath of terrain, making it appear that this terrain is the same temperature as that of the sides of these trees. Though the parallax effect causes this same "lay over" of trees elsewhere in the imagery, those instances are of much less concern since the terrain behind the trees rises uphill (therefore the area of terrain they are projected onto is much smaller). Ideally, regions of the image where the land surface is not visible (such as behind large trees) should be masked out like was done for the hill slopes hidden from view, and/or a digital surface model should be used for projection (one that includes the "surface" of large vegetation). Otherwise, it would be sufficient that the paper should at least mention this issue in the images, and call out the one particular region where the effect is most significant. Users of the imagery data can then mask out this region themselves. (And a related minor note, the road passes this same location as this stand of trees such that if an image was taken just as a car was driving by, the projection stretches the car across the terrain creating a narrow streak of very warm temperature in the image. See attached Figure 2. This is understandably not avoidable, but would be worth mentioning in the paper as an artifact of the image projection.)

We acknowledge the presence of important strikes pointed out by the reviewer. To compensate, we add the following paragraph to the Discussion section to inform dataset users: "Dataset users should account that the camera FOV contains not only snow but also rocks, trees, a road and wood fences. Moreover, during the orthorectification process, some vertical features such as tall trees or wood fences are projected over long distances on the DEM behind their actual position because of the oblique angle at which the camera is installed. This is always the case behind the snow fences in the central part of the image and can happen when cars are driving on the road in the camera FOV (Fig. 2)".

**Minor comments**

Lastly, I have one minor comment: Section 4.2 describes quantifying the transmissivity of the germanium window of the TIR camera housing. Are there specifications from the manufacturer of this window that these results can be compared to?

The transmittivity value of the camera window supplied by the manufacturer is 0.92, while we found 0.95. We recommend to carry on an independent assessment of the camera window properties for three reasons. First, no reflectivity or emissivity values were supplied. Second, the transmittivity supplied by the manufacturer is lower than the one of most Germanium windows with anti-reflection coating (0.95-0.99) available in commerce. Third, the anti-reflection coating of the window is made out of polymers that can be damaged by atmospheric UV radiation. We therefore also recommend to determine theses properties periodically. To make this aspect clear, we modify L167 as follows: "The manufacturer supplied a transmittivity value of 0.92 but no reflectivity or emissivity values. Moreover, we found this transmittivity to be lower than most Germanium windows with anti-reflection coating (0.95-0.99) available in commerce. Also, atmospheric UV radiation can damage the anti-reflection coating in time and cause these parameters to change. Transmissivity, reflectivity and emissivity ($t$, $r$ and $e$) of the camera's window were thus estimated experimentally to provide a correction".

Finally, although not requested by the reviewers, two figures were added to the manuscript to highlight the utility of the dataset. The first is a full timeseries of the snow surface temperature measured at AWS-South from between December 2022 and May 2023, compared with the TIR radiometer. The second illustrates four examples of spatial patterns of the snow surface temperature under different atmospheric conditions: a night with clear sky, a day with clear sky, a cloudy day and a melting day. To implement these figures, L376 was modified as follows: "The full timeseries are shown in Fig. 13. Finally, the images in Fig. 14, acquired under different atmospheric conditions, show the variability of the patterns of the surface temperature of snow caught by the TIR camera. Beside these examples, the dataset provides numerous other situations that deserve further exploration."